# A gene-centered *C. elegans* protein–DNA interaction network provides a framework for functional predictions

Juan I Fuxman Bass[1], Carles Pons[2], Lucie Kozlowski[1], John S Reece-Hoyes[1], Shaleen Shrestha[1], Amy D Holdorf[1], Akihiro Mori[1], Chad L Myers[2] & Albertha JM Walhout[1,*]

## Abstract

Transcription factors (TFs) play a central role in controlling spatiotemporal gene expression and the response to environmental cues. A comprehensive understanding of gene regulation requires integrating physical protein–DNA interactions (PDIs) with TF regulatory activity, expression patterns, and phenotypic data. Although great progress has been made in mapping PDIs using chromatin immunoprecipitation, these studies have only characterized ~10% of TFs in any metazoan species. The nematode *C. elegans* has been widely used to study gene regulation due to its compact genome with short regulatory sequences. Here, we delineated the largest gene-centered metazoan PDI network to date by examining interactions between 90% of *C. elegans* TFs and 15% of gene promoters. We used this network as a backbone to predict TF binding sites for 77 TFs, two-thirds of which are novel, as well as integrate gene expression, protein–protein interaction, and phenotypic data to predict regulatory and biological functions for multiple genes and TFs.

**Keywords** *C. elegans*; gene regulation; protein–DNA interaction network; transcription factors; yeast one-hybrid assays

**Subject Categories** Genome-Scale & Integrative Biology; Network Biology; Transcription

**Mol Syst Biol. (2016) 12: 884**

## Introduction

Accurate spatiotemporal gene expression is pivotal for development, the response to environmental stresses, and to maintain homeostasis. Specific gene expression patterns and levels are accomplished first and foremost by the action of regulatory transcription factors (TFs) that interact with non-coding DNA elements and activate or repress their target genes. When combined into gene regulatory network (GRN) models, physical and functional protein–DNA interactions (PDIs) between TFs and their target genes can provide insights into the regulation of gene expression at a systems level.

Gene regulatory networks have been studied at a small or medium scale in different metazoan organisms including nematodes, sea urchins, fruit flies, and mammals. For instance, GRNs regulating development in the sea urchin *Strongylocentrotus purpuratus* embryo have been delineated based on spatiotemporal expression patterns of TFs and signaling molecules, ChIP-seq data, and gene knockdown (reviewed in Martik *et al*, 2016). Similarly, developmental GRNs have been studied in *Drosophila melanogaster* using genetic perturbations, expression profiling, transgenic reporters, ChIP-seq data, and mathematical modeling (reviewed in Wilczynski & Furlong, 2010). More recently, massively parallel reporter assays have been used for an in-depth study of the function of regulatory sequences in cell lines, mouse tissues, yeast, and bacteria (reviewed in White, 2015).

The comprehensive large-scale experimental mapping of metazoan GRNs is a daunting task. The genomes of multicellular organisms harbor thousands of genes and regulatory elements such as promoters and enhancers, as well as large repertoires of TFs— usually 5–7% of protein-coding genes encode TFs. For instance, the human genome contains ~20,000 protein-coding genes, 1,434 of which encode TFs (Vaquerizas *et al*, 2009; Reece-Hoyes *et al*, 2011a). In addition, ~70,000 promoters and ~400,000 enhancers have been identified in the human genome (Bernstein *et al*, 2012). Thus, the comprehensive PDI mapping requires interrogating millions of element–TF combinations.

Over the last decade or so, many efforts have focused on determining consensus TF binding sites with the goal of being able to predict TF binding for a given genome. Protein binding microarray (PBM) and SELEX assays have been applied to a variety of TFs from many organisms (Badis *et al*, 2009; Grove *et al*, 2009; Jolma *et al*, 2010; Weirauch *et al*, 2014; Narasimhan *et al*, 2015; Nitta *et al*, 2015). While useful for understanding the detailed mechanisms of TF–DNA recognition, it has become clear that the presence of a TF

1 Program in Systems Biology and Program in Molecular Medicine, University of Massachusetts Medical School, Worcester, MA, USA.
2 Department of Computer Science and Engineering, University of Minnesota–Twin Cities, Minneapolis, MN,USA
 *Corresponding author. Tel: +1 508 856 4364; E-mail: marian.walhout@umassmed.edu

binding site in regulatory elements such as promoters and enhancers, or elsewhere in the genome, is a rather poor predictor of *in vivo* binding (Won *et al*, 2010; Li *et al*, 2011; Pique-Regi *et al*, 2011). Therefore, complementary methods that actually detect TF binding to larger genomic elements, either *in vivo* or by heterologous approaches, need to be applied as well.

The most widely used method for the identification of TF binding to a genome of interest is chromatin immunoprecipitation (ChIP). ChIP can be referred to as a TF-centered or protein-to-DNA method, because it focuses on an individual TF of interest (Walhout, 2006). While enabling the genomewide determination of TF binding, ChIP is restricted to a subset of all TFs as anti-TF antibody availability is limited and because ChIP works best for broadly and highly expressed TFs (Fuxman Bass *et al*, 2015). In addition, even for a single TF that can be ChIPped well, the experiment needs to be repeated many times to cover different cell types or different developmental or physiological conditions. Altogether, only a minority of TFs (an estimated ~10%) have been subjected to ChIP in any metazoan system, despite large consortium efforts such as human ENCODE (Gerstein *et al*, 2012), mouse ENCODE (Cheng *et al*, 2014), and modENCODE (Araya *et al*, 2014; Slattery *et al*, 2014) projects.

Yeast one-hybrid (Y1H) assays provide a complementary method for the identification of PDIs in a heterologous system in the milieu of the yeast nucleus (Arda & Walhout, 2010; Walhout, 2011). Y1H assays are gene-centered, or DNA-to-protein, because they start with a regulatory element of interest, such as a gene promoter or enhancer, and identify, in a single experiment, the entire repertoire of TFs that can bind that element (Deplancke *et al*, 2004, 2006; Vermeirssen *et al*, 2007). We have developed "enhanced" Y1H assays (eY1H) for use in high-throughput settings by introducing a robotic platform of 1,536 colonies expressing individual TFs in quadruplicate (i.e., assaying up to 384 TFs per plate) (Reece-Hoyes *et al*, 2011a,b). While eY1H assays do not interrogate TF binding in its natural setting, they have certain advantages when compared to ChIP, such as testing all possible TFs in a single experiment, including those not amenable to ChIP. However, there are also distinct disadvantages of eY1H assays, including the lack of detection of heterodimers or other more complex interactions involving multiple TFs (Walhout, 2011). Nonetheless, it is important to note that previous studies have shown a significant overlap between interactions detected by both methods (Brady *et al*, 2011; Reece-Hoyes *et al*, 2013; Fuxman Bass *et al*, 2015). Further, we have found that eY1H interactions could be validated in animals harboring transcriptional fusion reporter constructs fed with bacteria expressing RNAi clones against different TFs (MacNeil *et al*, 2015).

The nematode *Caenorhabditis elegans* has been used extensively as a model system to gain insights into the structure, function, and evolution of TF networks (Denver *et al*, 2005; Deplancke *et al*, 2006; Martinez *et al*, 2008b; Arda *et al*, 2010; Gerstein *et al*, 2010; Reece-Hoyes *et al*, 2013; Fuxman Bass *et al*, 2014). The *C. elegans* genome harbors 20,447 protein-coding genes (ensembl.org, Assembly WBcel235), 941 of which encode predicted TFs (Reece-Hoyes *et al*, 2005, 2011b) (this study). *C. elegans* has a compact genome with short intergenic regions and short introns (The *C. elegans* Sequencing Consortium, 1998). Therefore, most gene regulation likely occurs through proximal gene promoters. Indeed, the majority of TF binding events determined by ChIP reside within the first 500 bp of proximal gene promoters (Niu *et al*, 2011). Multiple

genomic resources have been generated for *C. elegans*, including Gateway-compatible ORFeome and promoterome resources that contain clones for more than 12,000 open reading frames (ORFs) (Reboul *et al*, 2003; Lamesch *et al*, 2004) and more than 5,000 promoters (Dupuy *et al*, 2004), respectively. In addition, we have generated a comprehensive clone resource of mostly full-length TFs that includes a set of unconventional DNA-binding proteins we discovered previously (i.e., proteins that can bind DNA but that lack a recognizable DNA-binding domain) (Deplancke *et al*, 2006). These clones can be used for physical GRN mapping by Y1H assays (Vermeirssen *et al*, 2007; Reece-Hoyes *et al*, 2011b), and for *in vivo* TF activity mapping by RNAi (MacNeil *et al*, 2015).

In addition to mapping physical interactions between TFs and genomic loci, a comprehensive characterization of GRNs requires the assessment of additional functional TF parameters. These include the regulatory function of each TF, which can be a repressor, activator, or bifunctional regulator of target gene expression. Furthermore, the biological function in the context of development, homeostasis, and physiology needs to be explored for each TF. Finally, it is important to uncover which TFs share targets and function redundantly in the control of gene expression.

Here, we use eY1H to examine ~2.8 million pairwise interactions between 3,373 *C. elegans* promoters and 837 full-length TFs, representing the largest gene-centered metazoan PDI screen to date. We combine the interactions with published eY1H data for TF-encoding gene promoters (Reece-Hoyes *et al*, 2013) to obtain a PDI network of 21,714 interactions between 2,576 genes and 366 TFs. By integrating the PDI network with gene expression and protein–protein interaction data, we provide predictions of the regulatory function for 170 TFs. In addition, we provide predictions of biological functions both for TFs and their target genes and confirm several of these predictions *in vivo*. Finally, we identify TFs that share a large proportion of eY1H targets, which serves as a blueprint to study redundancy and other epistatic relationships between TFs.

# Results

## A gene-centered *C. elegans* PDI network for 15% of protein-coding genes

A comprehensive study of *C. elegans* TF binding and function requires a high-throughput method that can interrogate multiple TFs in parallel under highly standardized laboratory conditions. We reasoned that using gene-centered eY1H assays to determine TF binding to a large set of promoters may provide a backbone for characterizing TF function when integrated with publicly available functional datasets.

In eY1H assays, a DNA region of interest (DNA bait) is cloned upstream of two reporter genes, *HIS3* and *LacZ*, and integrated into the yeast genome (Deplancke *et al*, 2004). TFs fused to the yeast Gal4p activation domain (preys) are introduced into the DNA bait strain by mating using a robotic platform (Reece-Hoyes *et al*, 2011b). To increase the confidence in the interactions detected, each TF is tested in quadruplicate (i.e., it occurs in quadrants on the TF array), and each DNA bait is tested in duplicate (i.e., with both reporters).

We primarily focused on available promoter clones from the promoterome resource (Fig 1A) (Dupuy *et al*, 2004). This collection comprises ~5,500 promoter regions of 0.3-2 kb of genes encoding for different molecular functions, and that are distributed across all *C. elegans* chromosomes. Promoters were transferred to the two Y1H bait Destination vectors by Gateway cloning (Walhout *et al*, 2000) and integrated into the yeast genome, successfully generating Y1H bait strains for 3,373 promoters corresponding to 3,364 genes. These strains were then used in eY1H assays to test pairwise interactions with 837 full-length *C. elegans* TFs (Fig 1B). Thus, in total we tested 2.8 million distinct TF-gene pairs, which represents the largest gene-centered metazoan PDI screen to date. The technical quality of the data was ensured by only considering interactions in which both eY1H reporters and at least two of the four colonies tested per TF scored positively (Reece-Hoyes *et al*, 2011b). In agreement with previous observations, all four colonies scored positively in a large majority of cases (~90%) (Reece-Hoyes *et al*, 2013; Fuxman Bass *et al*, 2015). We combined the resulting PDI data with those obtained for 678 TF-encoding gene promoters (Reece-Hoyes *et al*, 2013) to obtain a combined dataset of 4,051 promoters, covering 4,018 genes (Fig 1A). Altogether, we identified interacting TFs for 3,246 promoters corresponding to ~15% of *C. elegans* protein-coding genes (Dataset EV1). To obtain a high-quality PDI network, we removed promoters that conferred high or uneven background reporter gene expression in yeast. The final network contains 21,714 PDIs between 2,576 genes and 366 TFs (Fig 1A and B; Dataset EV1).

On average, each promoter in the network was bound by 8.4 TFs (median = 5), and each TF bound 2.3% of promoters tested (median = 0.4%). There was a large range in connectivity with promoters being bound by between one and 75 TFs, and a small number of TFs interacting with a very large proportion of promoters (Fig 1C). However, the majority of TFs bound only few promoters: 67% of the TFs detected bound fewer than 1% of promoters.

Does this PDI network represent interactions that are direct and that occur *in vivo*? To address this question, we tested the concurrence between eY1H interactions and PBM-derived TF binding sites (Narasimhan *et al*, 2015), as well as between the eY1H interactions and ChIP-seq data from the modENCODE consortium (Araya *et al*, 2014). First, we found a significant overlap between eY1H interactions and TF binding sites (Fig 1D). Further, we found that *de novo* motifs derived from eY1H data are overall similar to those determined by PBMs (Fig EV1; Dataset EV2). Importantly, we provide novel potential motifs for 52 TFs for which PBM data are not available (Dataset EV2). In addition to the concordance between eY1H and PBM data, we detected a weak but significant overlap between eY1H interactions and ChIP data from the modENCODE consortium (Fig 1E) (Araya *et al*, 2014). Only considering TFs that occur in both datasets, we found that 20% of eY1H interactions were also detected by ChIP, which is similar to our previous observations (Reece-Hoyes *et al*, 2013; Fuxman Bass *et al*, 2015). eY1H interactions not detected by ChIP may be occurring only in a few cells in the animal and be under the detection limit of ChIP when whole animals are used, or may occur in stages/conditions that were not used in ChIP assays. PDIs detected by ChIP but not eY1H may either be eY1H false negatives or, perhaps more likely, may be indirect *in vivo* (Liang *et al*, 2014; Narasimhan *et al*, 2015). Overall, these analyses

show that eY1H assays can recapitulate physical interactions found by *in vivo* ChIP or predicted based on *in vitro* PBMs.

So far, we have shown that the eY1H PDI network represents high-quality physical interactions. However, it has become clear that physical interactions do not always confer a regulatory consequence (Kemmeren *et al*, 2014; MacNeil *et al*, 2015). Therefore, we next asked whether the network as a whole represents regulatory TF-gene relationships. To do so, we reasoned that TFs would tend to be expressed in the same tissue(s) as their target genes, especially when the TF activates the expression of that gene. To test this, we integrated the eY1H PDI network with previously reported spatiotemporal gene expression data from different tissues (intestine, neurons, pharyngeal muscle, body wall muscle, coelomocytes, hypodermis) during embryo and larval stages (Spencer *et al*, 2011). We found that the overlap in expression patterns is significantly higher for TF-gene pairs that interact in eY1H assays than for TF-gene pairs that do not interact (Fig 1F). This enrichment is more striking for activators, while repressors are significantly depleted for interactions with genes that have similar expression patterns across tissues. We next combined the network with a compendium of co-expression data obtained from 123 expression profiling datasets (Chikina *et al*, 2009; Reece-Hoyes *et al*, 2013) and found that genes with high TF profile similarity, that is, that share a large proportion of interacting TFs in eY1H assays, are more frequently co-expressed than those with low TF profile similarity (Fig 1G). Altogether, these analyses indicate that the eY1H network captures physical interactions that occur *in vivo* and that it globally conveys gene regulatory events.

### eY1H assays are complementary to ChIP and PBM assays

Our PDI network greatly expands the number of *C. elegans* TFs for which DNA binding data are available. We detected interactions for 409 TFs, 366 of which engage in high-confidence PDIs (Fig 1B), while ChIP-seq and PBM data are available for 87 and 181 TFs, respectively (Fig 2A) (Araya *et al*, 2014; Narasimhan *et al*, 2015). Importantly, we detected interactions for 205 TFs for which no ChIP-seq or PBM data were heretofore available (Fig 2A).

Some TFs may be more suitable for detection by particular PDI mapping methods than others. For instance, only 10% of the 268 *C. elegans* nuclear hormone receptors (NHRs) have been successfully assayed by PBM assays (Weirauch *et al*, 2014; Narasimhan *et al*, 2015), while three times as many were detected in eY1H assays (Fig 2B; Dataset EV3). Further, we detected PDIs for TFs from all major families, including some that have not been detected or tested by PBMs such as Myb-like domain TFs and ZF-CCCH TFs, respectively. However, eY1H assays cannot detect complex interactions involving multiple TFs, while PBM and SELEX assays can (Grove *et al*, 2009; Jolma *et al*, 2015). We have previously shown that eY1H assays can detect human TFs that are expressed at a range of levels *in vivo*, while TFs successfully assayed by ChIP-seq are generally highly expressed (Fuxman Bass *et al*, 2015). Here, we confirm this finding in another species: *C. elegans* TFs detected by eY1H assays are less biased toward highly and broadly expressed TFs compared to those assayed by ChIP-seq (Fig 2C and D). Overall, these data show that eY1H assays expand our ability to probe PDIs and are complementary to *in vivo* ChIP and *in vitro* PBM assays.

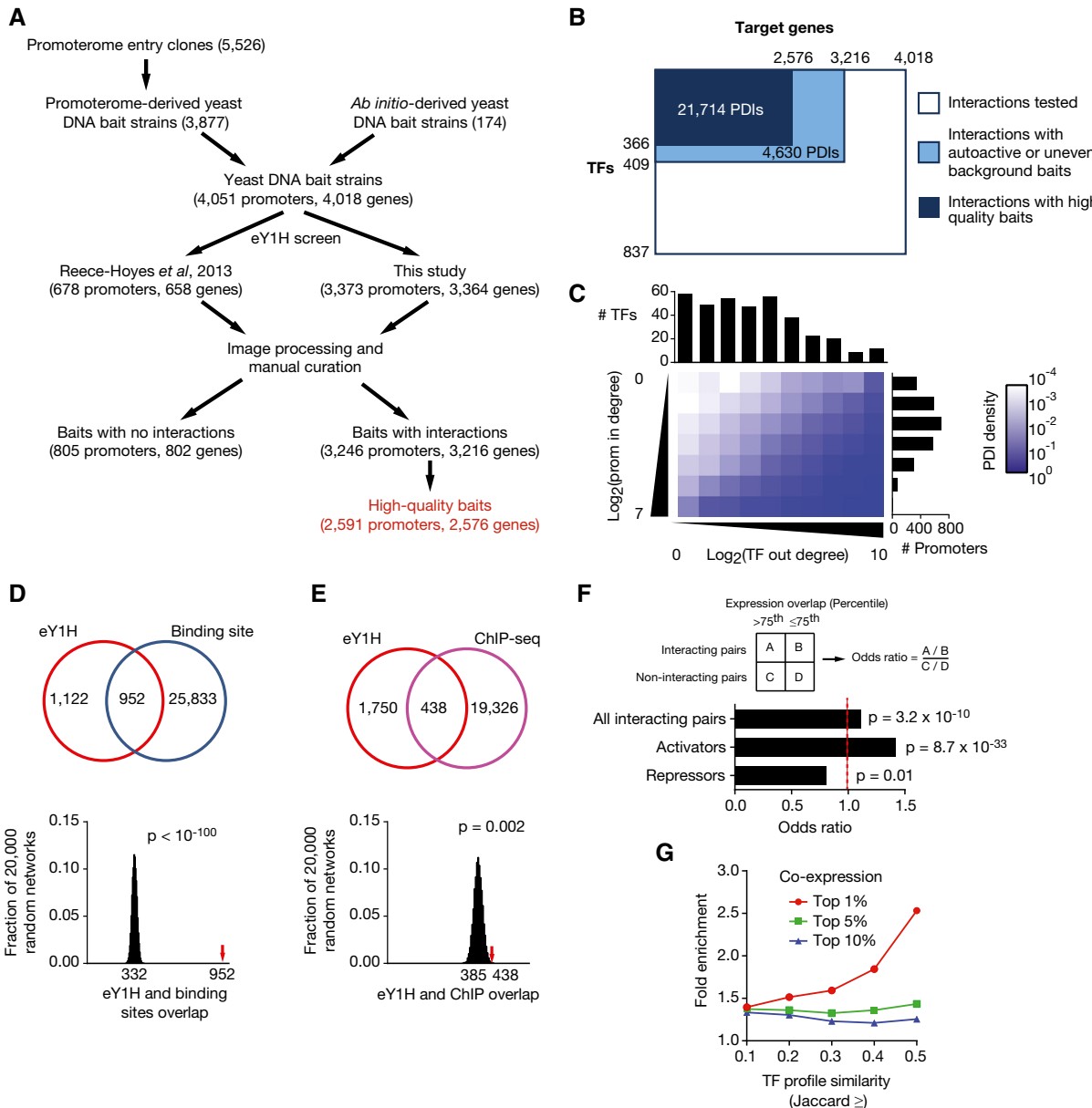

**Figure 1.  A *C. elegans* gene-centered PDI network for 15% of all genes.**

A   Flow diagram for yeast DNA bait generation and screening in eY1H assays.

B   PDI test space. Promoters of 4,018 *C. elegans* genes were screened using eY1H assays against an array of 837 TFs (~90% of 941). Highly auto-active (or uneven background) DNA baits and those that did not produce any interactions were removed (805). Interactions for DNA baits with moderate auto-activity or uneven background are included in Dataset EV1, but were excluded from the network (light blue). The final network contains 21,714 PDIs between 2,576 genes and 366 TFs.

C   Matrix representation of the PDI network. TF out-degree (*x*-axis), that is, the number of promoters a TF binds, and promoter in-degree (*y*-axis), that is, the number of TFs that bind a promoter, were binned in log2 scale. Each box in the matrix represents the density of PDIs calculated as the number of PDIs in the bin divided by the number of PDIs tested (i.e., the number of TFs multiplied by the number of promoters in that bin). Histograms on top and right of the matrix represent the number of TFs and promoters in each bin respectively.

D, E   eY1H interactions significantly overlap with the occurrence of known TF binding sites (D) or ChIP-seq interactions (E). The Venn diagrams (top) illustrate the number of overlapping interactions. The eY1H PDI network was randomized 20,000 times by edge switching, and the overlap for each randomized network was calculated (bottom). The numbers below the histogram peaks indicate the average overlap in the randomized networks. The red arrows indicate the observed overlap in the real eY1H network. Statistical significance was calculated from z-score values assuming a normal distribution for the randomized networks.

F   Overlap between spatiotemporal expression patterns between TFs and their eY1H target genes. The fraction of TF-gene pairs with an expression overlap above the 75th percentile was compared between interacting and non-interacting pairs. The same analysis was performed for known activators and repressors. Statistical significance was determined by chi-square test.

G   Co-expression between genes bound by similar sets of TFs in a compendium of 123 expression profiling datasets. The enrichment of top 1, 5, and 10% co-expressed gene pairs was determined for gene pairs with a TF profile similarity above the indicated Jaccard value. Only promoters bound by three or more TFs, and TFs that bind < 5% of promoters were considered in this analysis. *P* < 0.01 for all data points by Fisher's exact test.

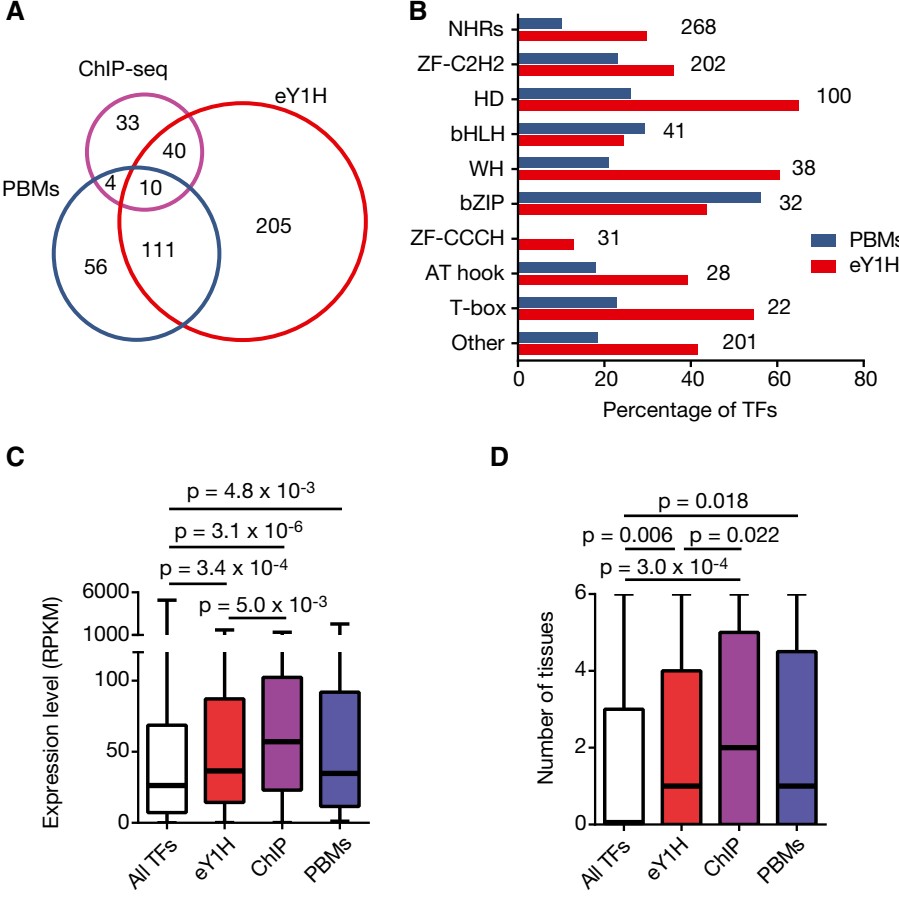

**Figure 2.  Comparison between TFs detected by different PDI mapping methods.**

A  Venn diagram depicting the overlap between the TFs detected by eY1H, ChIP-seq, and PBM assays.
B  Percentage of TFs from major TF families detected by eY1H and PBMs. Numbers on the right indicate the number of TFs corresponding to each TF family.
C  Boxplots indicating the distribution of maximum expression levels across development for TFs detected by eY1H, ChIP, or PBMs, or for all TFs.
D  Boxplots indicating the distribution of the number of tissues in which a TF is expressed in embryos for TFs detected by eY1H, ChIP, or PBMs or for all TFs.

Data information: In (C and D), each box spans from the first to the third quartile, the horizontal lines inside the boxes indicate the median value, and the whiskers indicate minimum and maximum values. Statistical significance determined by Mann–Whitney *U*-tests.

## Predicting activators and repressors

We reasoned that our large PDI network may provide an opportunity to generate and test predictions about the regulatory function of TFs, *that is,* whether they activate or repress gene expression. For the majority of *C. elegans* TFs, it is not yet known whether they are activators, repressors, or whether they both activate and repress transcription, depending on the cellular context (bifunctional TFs). The regulatory function of a TF can be determined by measuring gene expression changes caused by TF perturbation. We hypothesized that it may also be possible to infer the regulatory function of a TF based on the co-expression between the TF and its target genes across multiple expression profiling datasets. A positive correlation would result in the prediction that the TF is a transcriptional activator, while a negative correlation would suggest that the TF represses target gene expression (Fig 3A). Of course, such predictions can only be derived for TFs with a sufficiently high number of bound targets in eYH assays. Therefore, we focused on the 153 TFs that bind to the promoters of 10 or more

genes in eY1H assays and for which expression profiling data were available in more than 25 out of the 101 high-quality expression profiling datasets that were used in this analysis. For each TF, we compared the co-expression scores with its eY1H targets to the co-expression scores with its non-targets (Fig 3B). This analysis led to functional predictions for 44 TFs: 28 predicted activators and 16 predicted repressors. This number is significantly more than predictions based on random permutations of the co-expression scores for each TF (Fig 3C–E). Similar results were obtained when different cutoffs were selected for the number of promoters bound by a TF (Fig 3B).

One may expect that transcriptional activators are expressed in an overlapping set of tissues and developmental stages as their targets genes, while this is expected to be less common for repressors. Indeed, the majority of predicted activators have more similar expression patterns across tissues with their targets than with their non-targets, whereas repressors show the opposite relationship (Fig 3F). These observations support our overall functional TF annotations.

    

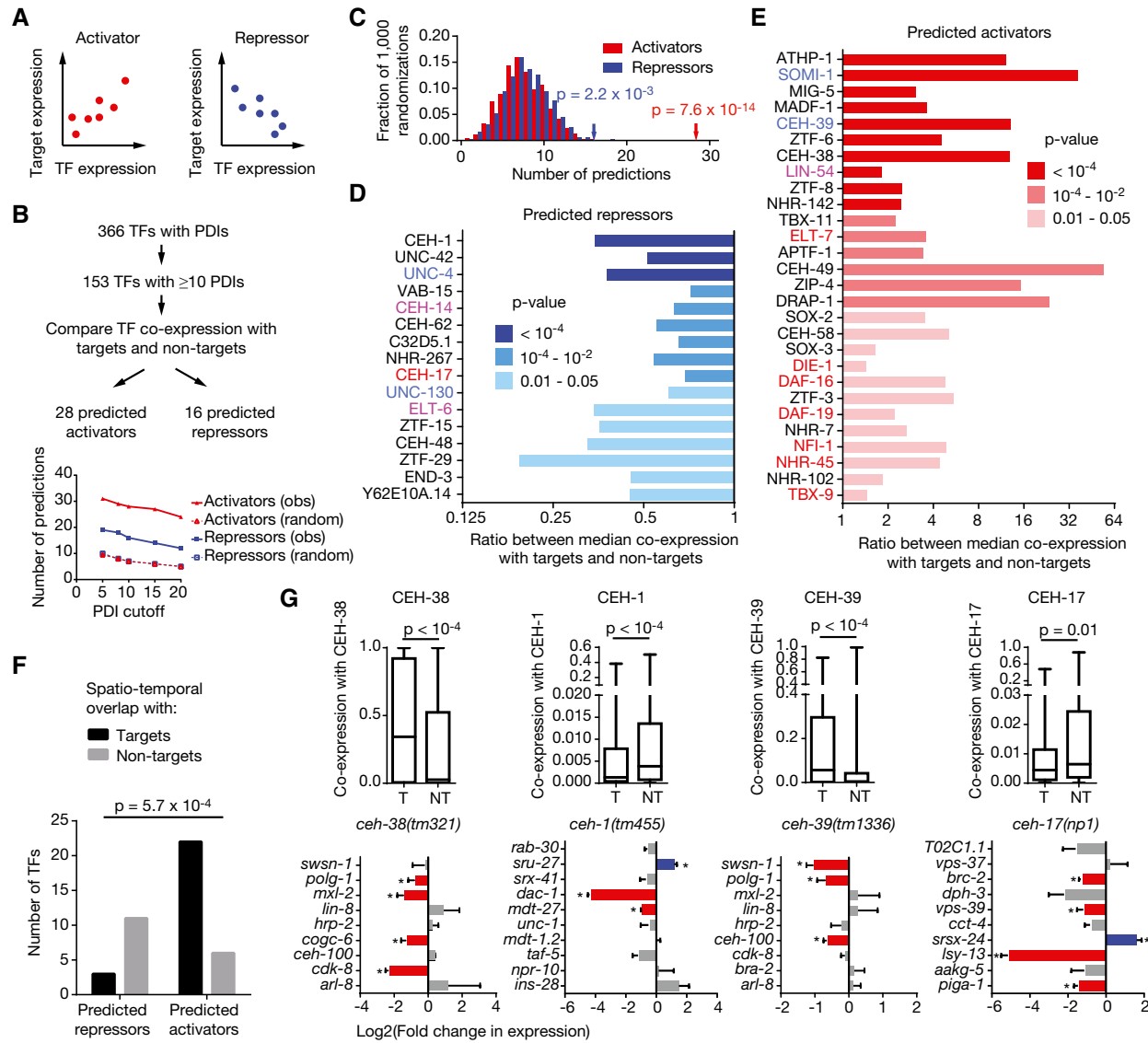

**Figure 3. Regulatory function prediction based on eY1H PDI network.**

A    Cartoon illustrating hypothetical co-expression between an activator or a repressor and its targets. Each dot represents the expression levels of a TF and a target gene in a single sample of an expression profiling dataset.

B    Flowchart for TF regulatory function predictions (top). The plot (bottom) indicates the number of activator and repressor predictions observed in the PDI network (obs) and the average number of predictions in 100 randomizations of the network (random), for TFs engaging in greater or equal number of PDIs than the indicated in the *x*-axis (PDI cutoff).

C    For each TF, the co-expression values with the eY1H genes were randomized 1,000 times and the number of activator and repressor predictions was calculated using *P*-value threshold of 0.05. The histograms represent the distribution of the number of predictions in the randomized networks. The red (activators) and blue (repressors) arrows indicate the observed number of predictions in the real network. Statistical significance was calculated from z-score values assuming normal distribution for the randomized datasets.

D, E  Predicted repressors (D) and activators (E). Co-expression across 123 expression profiling datasets was determined for all TF-gene pairs. Then, for each TF the ratio between the median co-expression with its eY1H targets and the median co-expression with its eY1H non-targets was determined. Shades of blue (D, repressors) and red (E, activators) indicate significance determined by Mann–Whitney *U*-tests. Predictions are provided for TFs with *P* < 0.05. TF names in blue represent known repressors, in red known activators, and in purple known bifunctional TFs.

F    Overlap between spatiotemporal expression patterns of TFs and their eY1H target genes. The number of TFs that are more frequently co-expressed with its targets and non-targets (with an expression correlation above the 75[th] percentile) is plotted for predicted activators and repressors. Statistical significance was determined by Fisher's exact test.

G    Validation of functional predictions for CEH-39, CEH-38, CEH-17 and CEH-1. TF co-expression with its eY1H targets (T) and non-targets (NT) across 123 expression profiling datasets (top). Each box spans from the first to the third quartile, the horizontal lines inside the boxes indicate the median value and the whiskers indicate minimum and maximum values. Statistical significance determined by Mann–Whitney *U*-tests. qRT–PCR analysis of a subset of eY1H target genes in TF mutant backgrounds (bottom). Values indicate the log2 fold change compared to N2. Red bars indicate significantly downregulated genes, blue bars indicate significantly upregulated genes, and gray bars indicate genes with no significant change in expression. Error bars indicate the standard error of the mean in three biological repeats (three technical replicates each). *P < 0.05 vs. N2 by two-tailed paired Student's *t*-test.

A regulatory function had been previously reported for 15 of the 44 TFs for which we provide functional predictions based on the eY1H network (Fig 3D and E; Dataset EV4). When we compared our predictions to these reported functions, we observed full agreement for nine TFs (60%) and an opposite prediction for three (20%). The remaining three (20%) involved bifunctional TFs. Overall, this level of agreement is encouraging because the reported functions are mostly based on one or few regulatory interactions. For instance, the reported regulatory functions of CEH-39 and SOMI-1, which are opposite to our predictions, are based on their effect in regulating one (*xol-1*) or two (*let-60* and *lin-14*) genes, respectively (Gladden & Meyer, 2007; Hayes *et al*, 2011).

To further test the regulatory effects of TFs *in vivo*, we measured eY1H target gene expression changes in mutant animals for four TFs: two with a previously unknown regulatory function (CEH-38 and CEH-1), and two with a predicted regulatory function opposite to what was reported previously (CEH-39 and CEH-17) (Gladden & Meyer, 2007; Van Buskirk & Sternberg, 2010).

Our analysis predicted that CEH-38 is a transcriptional activator (Fig 3E). We compared the expression of nine out of 108 randomly selected CEH-38 eY1H targets in *ceh-38(tm321)* mutant vs. wild-type animals by qRT–PCR. In support of our prediction, we found that four of these genes exhibited reduced levels in the mutant relative to wild-type animals, while five remained unchanged (Fig 3G).

We predicted CEH-1 to be a repressor (Fig 3E). However, we observed both increased and decreased levels of eY1H targets in *ceh-1(tm455)* mutants, relative to wild-type animals (Fig 3G). This suggests that CEH-1 may be a bifunctional TF that can both activate and repress gene expression *in vivo*.

We predicted CEH-39 to be an activator (Fig 3E); however, it has been associated with transcriptional repression based on its ability to bind to the regulatory regions of *xol-1* and inhibit its expression *in vivo* (Gladden & Meyer, 2007; Farboud *et al*, 2013) (Fig 3G). We evaluated the expression of nine CEH-39 eY1H targets in *ceh-39(tm1336)* mutants and found that three of nine eY1H targets exhibit reduced expression relative to wild-type animals, while six were unchanged (Fig 3G). This finding suggests that CEH-39 may actually function mostly as an activator, consistent with our prediction.

We predicted CEH-17 to be a repressor, which is inconsistent with its reported ability to induce gene expression in the ALA neuron (Fig 3G) (Deplancke *et al*, 2004; Van Buskirk & Sternberg, 2010). We evaluated the expression of ten CEH-17 eY1H targets in *ceh-17(np1)* mutant animals and found that it can both activate and repress gene expression: Four genes are reduced in expression and one is increased (Fig 3G). Overall, these observations indicate that eY1H data can be used to predict the regulatory functions of TFs, although predictions may be more accurate for activators. Moreover, the qRT–PCR validation data show that at least for the TFs tested, between 30 and 50% of the eY1H interactions confer a regulatory effect *in vivo*. It is likely that this number increases when more stages/conditions are examined.

## TF regulatory function prediction based on protein–protein interactions between TFs and cofactors

Transcriptional activation is mediated through interactions between TFs and co-activators and/or the transcriptional machinery, whereas repression is often mediated by interactions with co-repressors. We compared our predictions with previously delineated protein–protein interactions between TFs and cofactors (Reece-Hoyes *et al*, 2013). TFs that we predict to function as activators interact more frequently with co-activators such as mediator subunits and TBP-associated factors (TAFs) (Figs 4A and B and EV2), while predicted repressors interact more frequently with co-repressors such as Groucho (UNC-37) and the histone deacetylases HDAC-11 and SIN-3 (Figs 4A and B and EV2). These findings are similar to what is observed with published activators and repressors (Fig 4C) and validate our functional predictions.

Given the observed correlation between TF regulatory function and protein–protein interactions with co-activators and co-repressors, we hypothesized that TF–cofactor interactions could also provide predictions for the regulatory role of uncharacterized TFs, even those with few or no eY1H interactions. For instance, one could predict that a TF is an activator if it only interacts with co-activators and that it is a repressor if it only interacts with co-repressors, while a bifunctional TF may interact with both. Solely based on the protein–protein interaction data (Reece-Hoyes *et al*, 2013), we identified 62 predicted activators, 50 predicted repressors, and 30 bifunctional TFs (Fig 4D). These predictions are overall consistent with TF functions derived experimentally either in the literature or in this study ($P = 0.009$, activators vs. repressors), and also consistent with predictions based on co-expression between TFs and eY1H targets (Fig 4E–G; Dataset EV5). Importantly, this analysis led to predictions that could not be derived from the eY1H network and co-expression data due to a low number or lack of eY1H targets. For example, we predict NHR-71 to be an activator based on its interaction with the TBP-associated protein TAF-12. Although the regulatory function of NHR-71 was not predicted from the eY1H given that it only binds five promoters, *nhr-71* does have a median co-expression with its eY1H targets that is 12-fold higher than with its non-targets, supporting our prediction based on TF–cofactor interactions (Fig 4F). We also predicted NHR-31 and ZIP-2 to be activators based on their respective interactions with the histone acetyltransferase MYS-1 and the mediator subunit MDT-11, even though we did not detect any PDIs involving these TFs in eY1H assays. Consistent with our predictions, ZIP-2 was shown to be involved in the activation of genes involved in the response to the pathogen *P. aeruginosa* (McEwan *et al*, 2012), while NHR-31 activates genes encoding for subunits of the vacuolar ATPase (Hahn-Windgassen & Van Gilst, 2009). Overall, we provide functional predictions for 170 TFs based on the eY1H network and co-expression integration, and/or protein–protein interactions (Fig 4G and Dataset EV5). For 69 TFs, potential functional roles were derived from two independent sources: co-expression, protein–protein interactions, and/or experimentally derived data previously published or presented here (Fig 3G). In 88% of the cases, the evidence provided by the different sources is concordant, further illustrating the high quality of our predictions (Fig 4G).

## Novel target genes for the regulatory factor X TF DAF-19

The regulatory factor X (RFX) DAF-19 activates the expression of genes encoding for ciliary structures during the development of sensory neurons (Swoboda *et al*, 2000; Blacque *et al*, 2005). Out

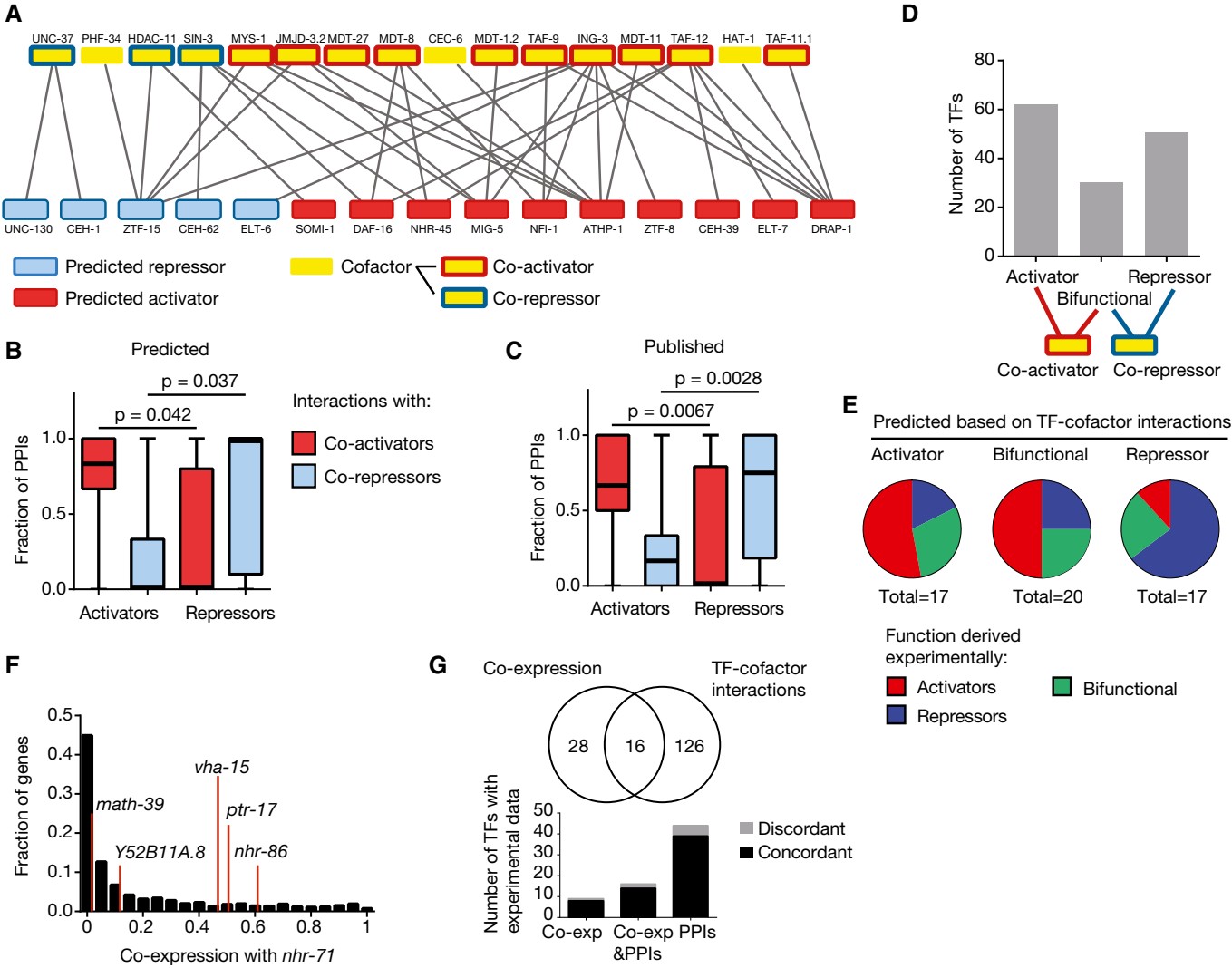

**Figure 4.  Regulatory function predictions based on TF–cofactor protein–protein interactions.**

A    TF–cofactor protein–protein interaction network from Reece-Hoyes *et al* (2013) for predicted activators and repressors. Blue, predicted repressors; red, predicted activators; yellow, cofactors; blue outline, co-repressors; red outline, co-activators.

B, C    Relationship between TF and cofactor functions. The fraction of protein–protein interactions (PPIs) with co-activators and co-repressors was determined for each predicted (B) or published (C) activator and repressor. Each box spans from the first to the third quartile, the horizontal lines inside the boxes indicate the median value, and the whiskers indicate minimum and maximum values. Statistical significance determined by two-tailed unpaired Student's *t*-test.

D    TF functional predictions based on TF–cofactor interactions. TFs were classified as potential activators if they only interact with co-activators, as repressors if they only interact with co-repressors, and bifunctional if they interact with both.

E    Experimentally derived functions from the literature and this study are shown for predicted activators, repressors, or bifunctional TFs based on TF–cofactor interactions.

F    Distribution of co-expression scores between *nhr-71* and genes that are not targets in eY1H assays (black histogram) and eY1H targets (red lines).

G    Overlap between TF regulatory predictions based on integrated eY1H and co-expression data or TF–cofactor interactions, and experimentally derived data. Black bars indicate the number of predictions that are concordant with experimental data.

of the 35 DAF-19 eY1H targets, 10 (29%) exhibited reduced levels in *daf-19* mutant animals (Phirke *et al*, 2011), and 16 harbor binding motifs for DAF-19 in their promoters but were not found to be regulated by DAF-19 (Blacque *et al*, 2005; Narasimhan *et al*, 2015) (Fig 5A). To determine whether some regulatory interactions may have been missed in the *daf-19* mutant expression profiling dataset (Phirke *et al*, 2011), we measured the expression change of the eY1H targets of DAF-19 by qRT–PCR in threefold embryos of *daf-19 (m86);daf-12(sa204)* animals compared to *daf-12(sa204)* animals

(the *daf-12* mutation suppresses the dauer-constitutive phenotype of *daf-19* mutants) (Senti & Swoboda, 2008). We confirmed that all previously reported DAF-19 targets are downregulated in *daf-19* mutant animals (Fig 5B). Importantly, this is also in agreement with our prediction that DAF-19 is a transcriptional activator (Fig 3E). We also detected significant downregulation for nine genes that were missed in the previous study (Phirke *et al*, 2011). Interestingly, these include three genes (F11E6.3, *fbxb-69*, and Y22D7AL.16) that do not harbor DAF-19 motifs in their promoters. This observation

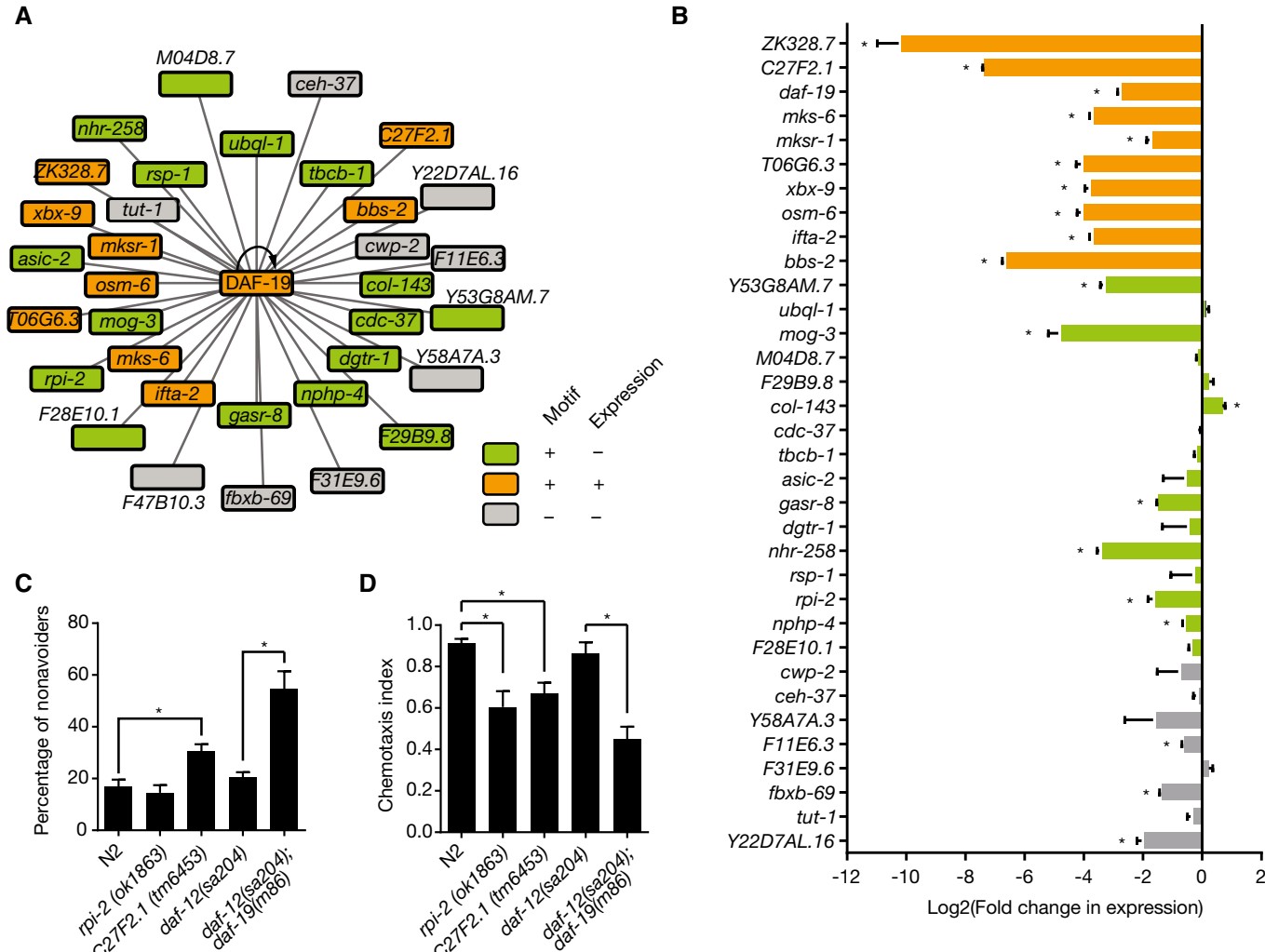

**Figure 5.  Novel target genes for the regulatory factor X TF DAF-19.**

A  eY1H targets of DAF-19. Green, promoters with DAF-19 TF binding sites; orange, genes regulated by DAF-19 with DAF-19 TF binding sites; gray, genes without DAF-19 TF binding sites nor regulated by DAF-19.

B  qRT–PCR analysis of DAF-19 targets in threefold embryos of a *daf-19(m86);daf-12(sa204)* mutant strain. Values were normalized to the expression in the *daf-12 (sa204)* strain (which was set to equal 0) and plotted as log2 fold change. Error bars indicate the standard error of the mean in three biological repeats. *$P < 0.05$ vs. *daf-12(sa204)* strain by two-tailed paired Student's *t*-test.

C  Osmotic avoidance assay for *rpi-2* and C27F2.1 mutants. A solution of 40% glycerol (15 μl) was added to the center of a 3.5-cm agar plate and let to dry. Then, ~100 animals were added to the center of the plate and incubated for 60 min at room temperature. The percentage of non-avoiders was calculated as the percentage of animals seeded within a radius of 0.8 mm. The average number of non-avoiders ± SEM from of six biological replicates (three technical replicates each) is plotted. *$P < 0.05$ vs. N2 by two-tailed paired Student's *t*-test.

D  Chemotaxis assay for *rpi-2* and C27F2.1 mutants. Chemotaxis to the chemoattractant isoamyl alcohol was measured as the difference between the number of animals attracted to isoamyl alcohol and those attracted to ethanol, divided by the total number of animals seeded after 2 h at room temperature. The average chemotaxis index ± SEM of three biological replicates (three technical replicates each) is plotted. *$P < 0.05$ vs. N2 by two-tailed paired Student's *t*-test.

suggests that DAF-19 may bind to additional, perhaps weaker, recognition sites in addition to its experimentally defined optimal motif.

*daf-19* mutants have a deficient differentiation of sensory neurons during embryonic development that is generally manifested phenotypically at later stages. For instance, *daf-19* mutant larvae and adults are chemotaxis-deficient, osmotic avoidance-defective, dauer-constitutive, and dye-filling-defective, among other phenotypes (Perkins *et al*, 1986; Malone & Thomas, 1994; Swoboda *et al*, 2000). Because DAF-19 is a TF, it is likely to exert its phenotypes

through the misregulation of one or more of its target genes. For instance, mutants in the DAF-19 targets *osm-6* and *nphp-4* are dauer-defective, chemotaxis-deficient, and osmotic avoidance-defective (*osm-6*) or exhibit dye-filling defects in sensory neurons (*nphp-4*) (www.wormbase.org). We reasoned that other targets of DAF-19 may also share some of the phenotypes of *daf-19* mutants. To test this hypothesis, we focused on the uncharacterized genes *rpi-2* and C27F2.1 for which mutant animals were available. We found that *rpi-2(ok1863)* mutant animals are chemotaxis-defective and that *C27F2.1(tm6453)* mutant animals are osmotic

avoidance- and chemotaxis-defective, although not to the same extent as *daf-19* mutant animals (Fig 5C and D). These data illustrate how the eY1H PDI network can be used to predict specific phenotypes for genes that are bound by a TF with a known function.

### Identification of novel functions for uncharacterized TFs

TFs often regulate the expression of functionally related genes, for instance, to coordinately respond to environmental or endogenous cues (Hsu *et al*, 2003; Araya *et al*, 2014). Therefore, we hypothesized that some TFs would preferentially bind to the promoters of genes belonging to particular biological process Gene Ontology terms (Fig 6A). Indeed, there are several known associations in the eY1H network. For example, eY1H targets of CEH-6, a TF involved in molting (Fraser *et al*, 2000), are enriched in the GO term "collagen and cuticulin-based cuticle development", and targets of CEH-45, a TF expressed in the spermatheca, are associated with the GO term "genitalia development". Altogether, we found 84 TF-GO association involving 27 TFs, which is significantly higher than what is found in randomized PDI networks (Fig 6B and Dataset EV6).

Many of the TF-GO associations were previously unknown. For instance, several NHRs are associated with the poorly defined GO term "response to organic substance". Manual inspection of the list of interactions involving these NHRs showed overrepresentation of phase I and II detoxifying enzymes from the cytochrome P450, UDP-glucuronosyltransferase, glutathione S-transferase, and dehydrogenase/reductase families. Indeed, NHRs are enriched in the collection of TFs that preferentially bind to the promoters of these genes compared to the overall frequency of NHR binding to promoters in the PDI network (Fig 6C and D). This suggests that NHRs may play an important role in detoxification in *C. elegans.*

The *C. elegans* genome encodes 268 NHRs, in contrast to the human genome, which only encodes 48 (Reece-Hoyes *et al*, 2005; Vaquerizas *et al*, 2009). It has been proposed that the expansion of the NHR family in *C. elegans* may be related to the need to respond to an ever-changing environment as related parasitic nematodes encode for 20–100 NHRs (Taubert *et al*, 2011). NHRs often have a ligand binding domain that can recognize endogenous or xenobiotic substances, which modulate their regulatory activity (Evans, 1988; Savas *et al*, 1999; Waxman, 1999). For instance, the ligand for the *C. elegans* NHR DAF-12 is dafachronic acid, which is produced endogenously during larval development. Upon binding of its ligand, DAF-12 activates transcription of its target genes thereby regulating metabolism and life history traits (Motola *et al*, 2006). Other proposed NHR ligands in *C. elegans* include the plant-derived xenobiotics chloroquine and colchicine, which work via NHR-8, although direct binding has not been shown (Lindblom *et al*, 2001). Similarly, NHR-176 is involved in the detoxification pathway of the fungicide and parasiticide thiabendazole, but it remains to be determined whether these compounds function directly as ligands (Jones *et al*, 2015). However, except for these few examples, the global repertoire of NHRs involved in the response to xenobiotics remains unknown.

We reasoned that if NHRs are broadly involved in the transcriptional response to xenobiotics, knocking down these NHRs would render the animals more sensitive to specific compounds. We tested this hypothesis using RNAi-mediated gene knockdown of 26 NHRs

that bound to the promoters of at least one detoxifying enzyme. Synchronized L1-arrested animals were fed bacteria expressing double-stranded RNA against these NHRs and treated with 16 compounds with different structures and origins (plant- and bacteria-derived, synthetic plaguicides, and inorganic cations) that are known to be toxic to *C. elegans* (Fig 6E; Dataset EV7). After 3–4 days, we screened for toxicity based on lethality, larval arrest, or slow growth on the xenobiotic-treated animals and compared them to untreated animals. *nhr-23* and *nhr-67* were excluded from the analyses as RNAi against these TFs leads to larval arrest in the absence of toxic compounds as previously reported (www.wormbase.org). In total, we observed 13 xenobiotic-NHR interactions involving seven compounds and nine NHRs (Fig 6F). We validated six of ten interactions tested in NHR mutant animals (Fig 6F–H). For instance, we found that *nhr-142(tm4401)* mutant animals are more sensitive to the insecticide/nematicide aldicarb and the herbicide/fungicide/nematicide dazomet compared to N2 (Fig 6G and H). Overall, these findings illustrate how eY1H data can help to identify the biological functions for uncharacterized TFs when integrated with target gene functional annotations.

### Redundancy and complexity in gene regulation

A comprehensive study of gene regulation entails not only identifying the regulatory and biological function of each TF but also the functional relationships between them. TFs can have different functional relationships depending on whether they bind to similar DNA sequences, are co-expressed, and have similar or opposite regulatory functions. For instance, co-expressed TFs that bind to similar DNA sequences and that both activate or both repress gene expression may act redundantly (i.e., one TF can compensate for the absence of the other) (Hollenhorst *et al*, 2007; Ow *et al*, 2008). There are several known instances of biological TF redundancy in *C. elegans*. For instance, the T-box TFs TBX-37 and TBX-38 are both expressed in the ABa descendant cells and are redundantly required for viability during early embryonic development (Neves & Priess, 2005). In the eY1H PDI network, these two TFs share 40 out of 92 target promoters (Fig 7A) suggesting that these proteins may function redundantly by regulating an overlapping set of genes. Of course, two TFs that bind similar DNA sequences can also have different functions in the animal, especially if they have different expression patterns and/or if they have opposite effects on gene expression (one is an activator and the other a repressor).

To globally uncover functional relationships between TFs, we leveraged the PDI network to identify TF pairs that bind overlapping sets of targets in eY1H assays and that, as we have previously shown, frequently bind to similar DNA sequences (Reece-Hoyes *et al*, 2013; Fuxman Bass *et al*, 2015). Such target sharing relationships were visualized in a TF association network, where two TFs are connected if their target profile similarity (i.e., the number of shared targets divided by the number of targets that bind either TF) is ≥ 0.2 (Figs 7B and EV3). As expected, paralogous TFs, which often have similar DNA-binding specificities, generally cluster together in this network (Grove *et al*, 2009; Reece-Hoyes *et al*, 2013; Weirauch *et al*, 2014; Fuxman Bass *et al*, 2015). An important question is whether connections in this TF association network globally represent redundant TF pairs. We hypothesized that TFs that are more connected in the association network would be less

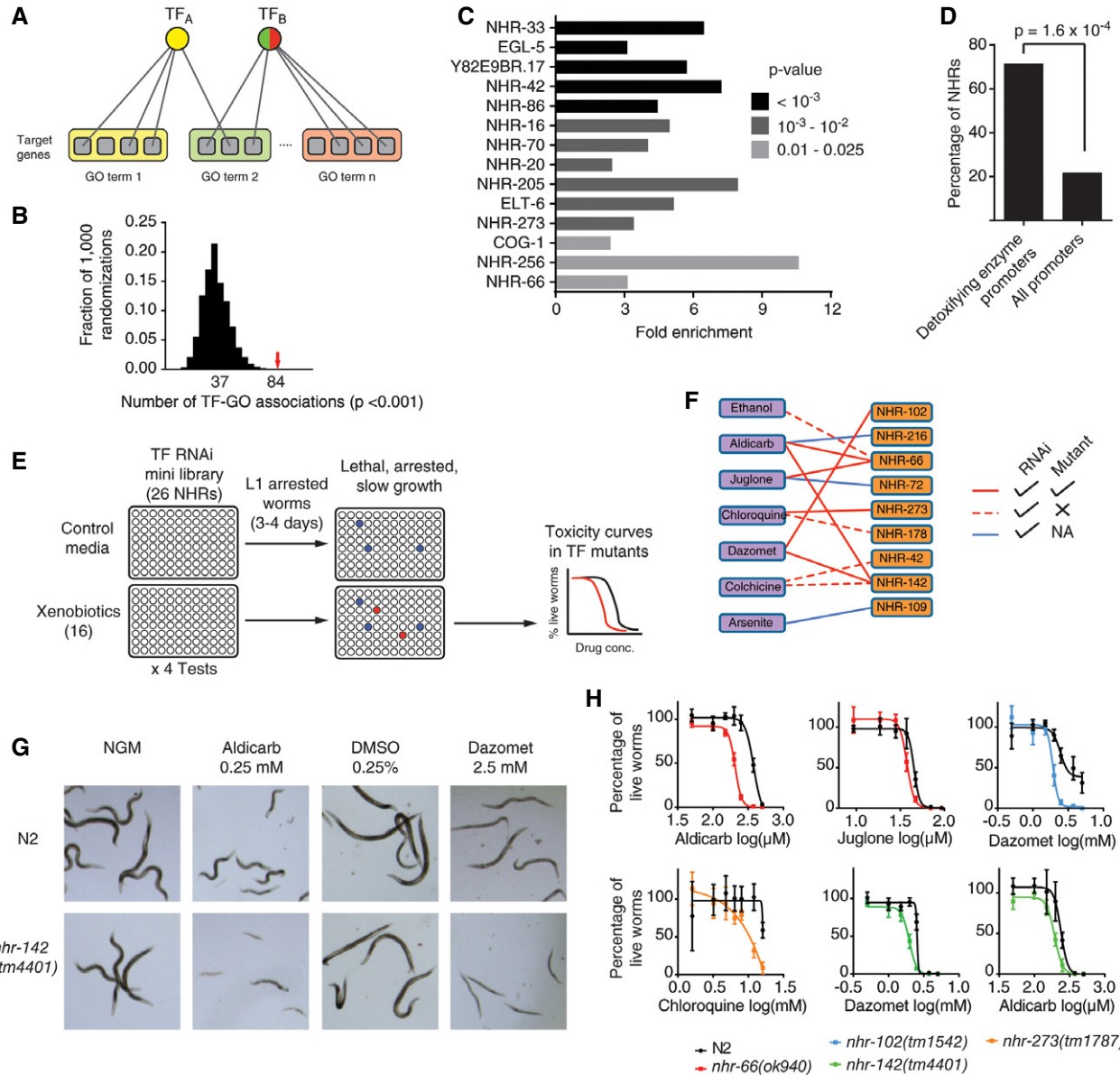

**Figure 6. Xenobiotic-NHR synthetic interactions.**

A   Schematic of TF-gene ontology (GO) association predictions. For each TF, the enrichment in binding to the promoters of genes belonging to different Biological Processes gene ontologies was determined. Edges indicate PDIs. Gray squares, genes; yellow, green, and red rectangles, Gene Ontology terms. TFs are colored based on the predicted TF-GO association.

B   The eY1H network was randomized 1,000 times by edge switching, and the number of TF-GO association predictions was calculated using $P < 0.001$. The numbers under the histogram peaks indicate the average number of predictions in the randomized networks. The red arrow indicates the observed number of predictions in the real network.

C   TFs enriched in binding to the promoters of detoxifying enzymes. Fold enrichment for TF binding to the promoters of CYP, GST, UGT, and SDR genes compared to all genes in the PDI network. Shades of gray indicate significance determined by Fisher's exact test.

D   Enrichment of NHRs binding to the promoters of detoxifying enzymes vs. all promoters in the PDI network. Statistical significance determined by proportion comparison test.

E   Schematics of RNAi experiments on animals treated with different drugs. L1-arrested N2 animals were added to plates seeded with TF RNAi bacterial clones and one of 16 drugs. After 3–4 days the animals were scored for toxicity phenotypes. TFs whose knockdown increases toxicity (red wells) were then tested using mutant strains using different drug concentrations.

F   Xenobiotic-NHR network. Purple rectangles, xenobiotics; orange rectangles, NHRs; solid red edges, interactions detected in RNAi and mutant experiments; dashed red edges, interactions detected in RNAi but not in mutant experiments; blue edges, interactions detected in RNAi experiments and not tested with mutant animals.

G   Aldicarb and dazomet sensitivity of *nhr-142(tm4401)* animals. L1-arrested N2 or *nhr-142(tm4401)* animals were treated with aldicarb (0.25 mM) or dazomet (2.5 mM) for 3 days and then examined for toxicity phenotypes.

H   Toxicity assays for mutants in NHRs. L1 animals from the indicated strains were treated with different drug concentrations and incubated for 3 days. The percentage of animals that reach the L3 stage ± SEM of four technical replicates is indicated. Representative experiments of three biological replicates.

    

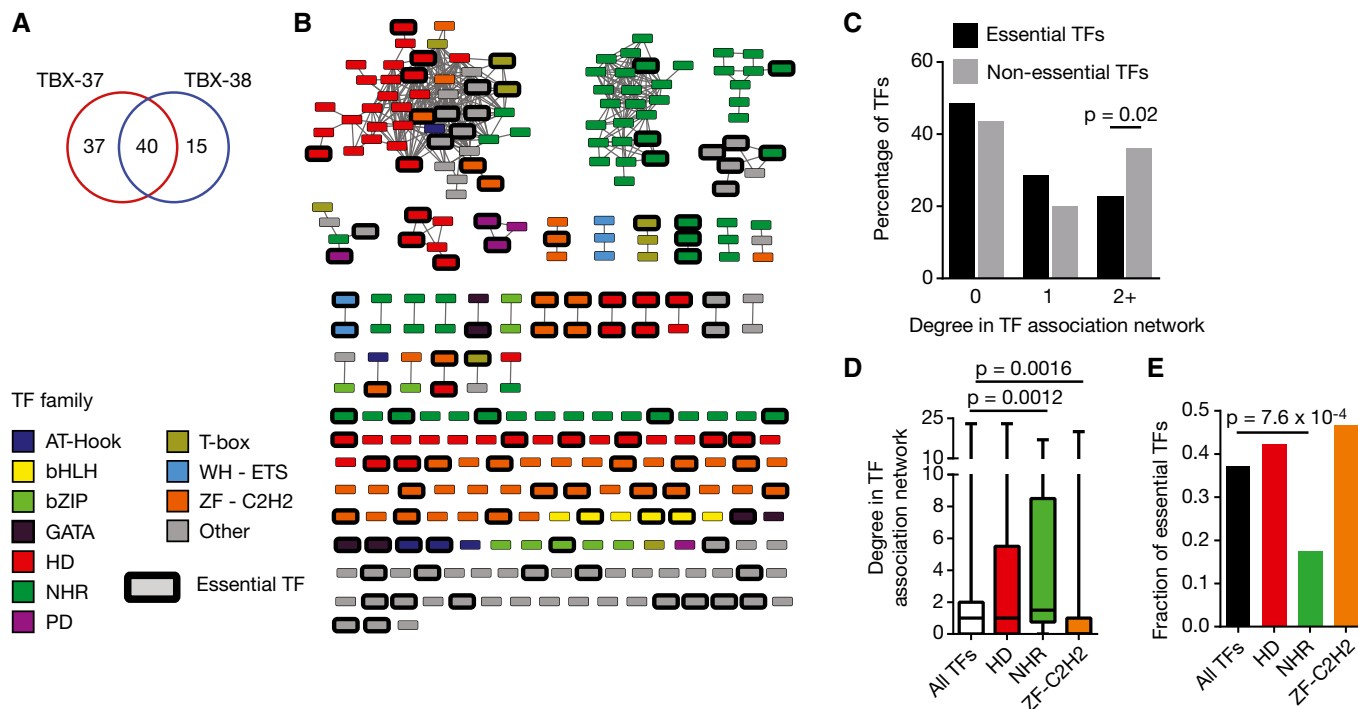

**Figure 7. TF redundancy.**

A Overlap in the number of eY1H targets between TBX-37 and TBX-38.

B TF association network. TFs are connected by an edge when they share eY1H target genes with a target profile similarity ≥ 0.2. Only TFs with degree ≥ 3 in the eY1H network are shown. Node color indicates TF families. Essential TFs are highlighted by a black outline. bHLH, basic helix-loop-helix; bZIP, basic leucine zipper domain; HD, homeodomain; NHR, nuclear hormone receptor; PD, paired domain; WH-ETS, winged helix E26 transformation-specific; ZF-C2H2, zinc finger C2H2.

C Relationship between essentiality and TF connectivity in the TF association network. The distribution of TF degree in the association network is plotted for essential and non-essential TFs. Statistical significance determined by proportion comparison test.

D Degree in the association network for TFs from different families. Each box spans from the first to the third quartile, the horizontal lines inside the boxes indicate the median value and the whiskers indicate minimum and maximum values. Statistical significance determined by Mann–Whitney *U*-tests.

E Fraction of essentiality for different TF families. Statistical significance determined by proportion comparison test.

frequently essential because loss of one such TF could be masked by another. Indeed, non-essential TFs are more highly connected in the association network than essential TFs, indicating that the PDI network globally captures TF redundancy (Fig 7C).

The connectivity in the association network is not uniform for all TF families. For instance, NHRs are generally more connected relative to all TFs (Fig 7D). This observation suggests that many NHRs bind to similar DNA sequences. In contrast, ZF-C2H2s are more isolated in the association network, indicating that they have more distinct interaction profiles (Fig 7D). This is consistent with the greater diversity in DNA-binding specificity for the ZF-C2H2 TF family (Jolma *et al*, 2013; Weirauch *et al*, 2014). The differential connectivity of NHR and ZF-C2H2 TFs in the association network negatively correlates with the fraction of essential TFs in these families (Fig 7E), suggesting that different TF families may have different redundancy potentials.

To explore the potential functional redundancy of a pair of TFs with a large number of shared eY1H targets, we focused on the close paralogs NHR-102 and NHR-142, which share 69 eY1H targets and have a high DNA-binding domain amino acid identity (59%) (Fig 8A). To determine whether these two TFs are redundant, we measured the expression of nine of their shared targets by qRT–PCR in wild-type, single-mutant, and double-mutant animals (Fig 8B).

Indeed, we did find evidence for functional redundancy between the TFs, but this was limited to one of their nine shared targets: NHR-102 and NHR-142 appear to redundantly repress their common target, *nhr-178*. Surprisingly, we observed that for three other targets among the nine tested, *cts-1, clec-42,* and *sru-12,* regulation by NHR-102 and NHR-142 was antagonistic. For these targets, the expression change in the double mutant was lower than the expected based on the expression in the single mutants. Given that these three target genes play a role in development, we wondered whether NHR-102 and NHR-142 may have antagonistic effects during development. Indeed, while the developmental rate of *nhr-102(tm1542)* animals is reduced compared to that of N2 and *nhr-142(tm4401)* animals, double-mutant animals display a similar developmental rate as N2 animals (Fig 8C). Overall, this suggests that even TFs that share a large number of physical interactions may have different functional relationships with each other with respect to each shared target gene, further illustrating the complexity in gene regulation.

# Discussion

This study presents the largest gene-centered PDI network to date. The high quality of the network is illustrated in multiple ways:

**Figure 8.  Complex epistatic relationships between NHR-102 and NHR-142.**

A   Overlap in the number of eY1H targets between NHR-102 and NHR-142.

B   qRT–PCR analysis of shared targets between NHR-102 and NHR-142 in single- and double-mutant animals. Values indicate the log2 fold change compared to N2. Error bars indicate the standard error of the mean in four biological repeats. *$P < 0.05$ vs. N2 by two-tailed paired Student's *t*-test. Redundant and antagonistic relationships were tested by two-way ANOVA with repeated measures. Significant interaction terms between target gene expression in the two mutant backgrounds ($P < 0.05$) are squared.

C   Developmental progression of animals at 45 h post-L1 synchronization of N2, *nhr-102(tm1542)*, *nhr-142(tm4401)*, and double mutants. Data are representative of four experiments.

(i) Each PDI was determined using two reporters and tested in quadruplicate, and only interactions identified in at least two replicates were considered positive; (ii) we found a significant overlap with other PDI datasets including the *in vitro* PBMs and the *in vivo* ChIP; and (iii) the network captures PDIs that are regulatory *in vivo*. Indeed, 30–60% of the PDIs tested in TF mutant animals resulted in changes in the expression of target genes. This result is similar to what has been previously reported for Y1H-derived interactions in *Arabidopsis thaliana* and *Drosophila melanogaster* (Brady *et al*, 2011; Hens *et al*, 2011). There are several reasons why we failed to detect a regulatory role for some PDIs detected by eY1H (Walhout, 2011). First, some PDIs identified in yeast may not occur *in vivo*. Second, some physical interactions, although occurring *in vivo*, may be neutral and have no regulatory consequence (MacNeil *et al*, 2015). Third, some regulatory interactions may have been missed as they were tested in a single stage and condition, and because expression changes were measured in whole animals losing tissue resolution. Finally, some regulatory interactions may have been missed due to redundancy with other TFs.

The PDI network presented here provides a backbone for integrating other large-scale datasets for making functional predictions about TFs. For instance, by integrating the PDI network with a compendium of expression profiling data we provide predictions for the regulatory functions of 44 TFs, which are largely consistent with published regulatory functions and those predicted based on TF–cofactor interactions. This number of predictions is encouraging, considering that there are several reasons that can explain lack of correlation between TF and target gene expression. First, our predictions are based on the assumption that there is a strong correlation between TF mRNA expression and TF activity. However, this may not be the case for many TFs whose regulatory function is modulated by ligand binding, posttranslational modifications or dimerization with other factors (Walhout, 2011). For instance, although we provide regulatory predictions for 29% of the TFs evaluated, we did so only for 15% of the NHRs tested, a family known to be modulated by ligands and that frequently forms heterodimers. Second, predictions for some TFs may be missed due to redundancy as changes in the expression of a single TF may not be sufficient to affect the expression of its target gene. Finally, given that predictions are based on the comparison between TF co-expression with targets and non-targets (higher or lower), we cannot identify bifunctional TFs using this approach. Therefore, TFs that can function as both activators and repressors would have been missed in this analysis.

It has recently been shown that ~40% of yeast TFs function as transcriptional repressors (Kemmeren *et al*, 2014). Consistent with this study, ~35% of our predictions based either on PDIs and co-expression, or based on TF–cofactor interactions, correspond to repressors. This finding suggests that the high percentage of repressors encoded by the yeast genome may be a feature common to other eukaryotes, including metazoans. In addition, predicted activators and repressors engage in a similar number of PDIs in the network, 4,055 and 3,573 respectively, supporting the hypothesis that, as in yeast, transcriptional repression may be a widespread mechanism of transcriptional regulation in *C. elegans*. Further, predicted activators and repressors are equally likely to be essential (38% of predicted activators and 44% of predicted repressors are

essential), suggesting that repression of gene expression is also important for viability.

By comparing TF connectivity in the PDI network we found that TFs that share a large proportion of targets, and are therefore likely to bind to similar DNA sequences, are enriched in redundant TFs, although other functional relationships are also possible. Further, two TFs can have either synergistic or antagonistic effects on gene expression depending on the target gene, as we have shown for NHR-102 and NHR-142. In addition, many TF pairs connected in the association network could have unrelated functions if they are expressed in different tissues/conditions/stages, and/or respond to different environmental cues as we have observed for NHR mutant animals exposed to different xenobiotics. Overall, these observations highlight the complexity in *C. elegans* gene regulation and the importance of integrating multiple functional datasets.

Although we have focused here on functional relationships between TFs with similar specificities, functional relationships can also be found between TFs that bind to different DNA sequences on the same promoter. While we have not explored these relationships here, the PDI network can serve as a first step to guide the study of how multiple signals from different TFs impinging on a gene promoter are integrated to regulate gene expression.

Ultimately, a comprehensive characterization of TF function will require the integration of multiple high- and low-throughput datasets. The PDI network presented in this study can serve as a backbone with which newly generated expression, protein–protein interactions, phenotypic and functional datasets can be overlaid which will result in broader and more accurate functional predictions.

## Materials and Methods

### *C. elegans* TFs list wTF3.0

We updated our previously published list of *C. elegans* TFs (Reece-Hoyes *et al*, 2011b). First, we removed nine pseudogenes and four genes whose status changed to dead gene according to WormBase version WS252 (Dataset EV3). Second, we removed 18 genes that have been characterized as having non-TF functions such as *dcp-66*, *ubxn-1,* and *irld-33,* which are cofactors, membrane proteins (e.g., *frpr-9* and *srab-2*), and a protease (*atg-4.1*). The remaining set of TFs was supplemented with 21 unconventional DNA-binding proteins (i.e., proteins that can bind DNA but that lack a recognizable DNA-binding domain (Deplancke *et al*, 2006), hereafter grouped together with the TFs) for which we detected PDIs in this study, three genes that have been re-annotated in WormBase, six proteins classified as TFs by other groups that bind DNA *in vitro* in PBM assays (Narasimhan *et al*, 2015) or SELEX (Mathelier *et al*, 2014), and five proteins newly classified as TFs in WormBase based on having a known DNA-binding domain (*maf-1*, *zip-9*, *madf-6*, *nhr-236*, and T10B5.10). This updated compendium of TF predictions contains 941 TFs and is referred to as wTF3.0 (Dataset EV3).

### eY1H assays

Enhanced yeast one-hybrid (eY1H) assays detect PDIs between a "DNA bait" (e.g., a gene promoter) and "TF preys". We used the *C. elegans* promoterome resource (Dupuy *et al*, 2004) and generated DNA bait strains for 4,051 *C. elegans* promoters, corresponding to 4,018 genes (Fig 1A) (Reece-Hoyes *et al*, 2011b). Briefly, promoters were Gateway cloned into two Y1H Destination vectors that contain *HIS3* and *LacZ* reporter genes (Walhout *et al*, 2000; Deplancke *et al*, 2004). The two resulting DNA bait::reporter constructs were then integrated into the yeast genome at fixed loci to generate "DNA bait strains" (Reece-Hoyes & Walhout, 2012). As prey we used an array of yeast strains individually expressing 837 *C. elegans* TFs fused to the yeast Gal4p activation domain (AD) (Reece-Hoyes *et al*, 2011b). DNA baits and TF preys were introduced into the same cell by mating. When a TF binds the DNA bait, the AD moiety activates reporter gene expression. *HIS3* expression allows the yeast to grow on media lacking histidine and containing 3-amino-triazole (3AT), a competitive inhibitor of the His3p enzyme, while *LacZ* expression is detected via the conversion of colorless X-gal into a blue compound (Reece-Hoyes *et al*, 2011b).

eY1H assays were performed using a Singer RoToR robot that manipulates yeast strains in a 1,536-colony format (Reece-Hoyes *et al*, 2011b). Images of readout plates lacking histidine and containing 3AT and X-gal were processed using the Mybrid web-tool to automatically detect positive interactions (Reece-Hoyes *et al*, 2013). Each interaction was tested in quadruplicate, and only those that scored positive at least twice were considered (90% of the PDIs detected were supported by all four colonies). The generated images were integrated with our published dataset for 678 TF-gene promoters (Reece-Hoyes *et al*, 2013), and interactions detected by Mybrid were subsequently manually curated to (i) eliminate false positives detected by Mybrid on readout plates with uneven background and (ii) include interactions that were missed by Mybrid because they occur next to very strong positives or involve baits that exhibit uneven or moderately high background reporter gene expression. We detected PDIs for 3,216 genes (80%) (Dataset EV8). Baits with intermediate background or uneven reporter expression were removed from further analysis, resulting in a high-quality PDI network comprising 21,714 interactions between 2,576 target genes and 366 TFs (Dataset EV1).

### Interaction profile similarity

Target profile similarity of two TFs, A and B, was defined using the Jaccard index as the number of eY1H baits bound by both A and B, divided by the number of eY1H baits that interact with either A or B (Fuxman Bass *et al*, 2013). Similarly, TF interaction profile was defined as the number of TFs bound to both gene promoters, X and Y, divided by the number of TFs that interact with either X or Y.

### Overlap between eY1H interactions and TF binding sites

Position weight matrices (PWMs) for *C. elegans* TFs were obtained from CisBP (Weirauch *et al*, 2014; Narasimhan *et al*, 2015). Only TFs that were detected by eY1H assays and PBM assays were considered (121 TFs). To determine the presence of candidate TF binding sites in a promoter based on matches to PWMs, we used the energy-based scoring scheme implemented in the BEEML software package (Zhao *et al*, 2009). BEEML provides a score between 0 and 0.5 that indicates how well a given k-mer matches a PWM (0.5 is a perfect match). A threshold of 0.09 was used as cutoff to assign a TF binding site (Reece-Hoyes *et al*, 2013). Only the proximal 500 bp

of the promoter was considered given that most *in vivo* PDIs occur in this region (Niu *et al*, 2011). TFs with predicted binding sites in more than 50% of promoter sequences were considered non-specific (45 TFs) and were removed from the analysis. Altogether, BEEML was used to generate a predicted PDI network between 76 TFs and 2,576 promoters. Multiple matches with scores above 0.09 between a PWM and a promoter fragment were considered a single interaction. To calculate the statistical significance of the overlap between the eY1H and the BEEML-predicted interactions, we compared the BEEML-predicted interactions with 20,000 randomizations of the eY1H network. Randomization was done by edge switching so that overall network topology as well as individual node degree was preserved (Martinez *et al*, 2008a).

### TF motif predictions based on eY1H data

Motif predictions were derived using CisFinder (Sharov & Ko, 2009). Briefly, for each TF, the 500 bp proximal promoter sequences of eY1H targets was used to identify motifs enriched compared to the sequences of promoters not bound in eY1H assays. Elementary motifs were obtained using an FDR of 0.05 and a motif enrichment > 2. Motifs were determined for the 77 TFs that have 50 or more eY1H targets in the PDI network to avoid obtaining low-quality motifs resulting from a low number of positive sequences (Dataset EV2). Statistical significance between the eY1H- and PBM-derived motifs was determined using the TOMTOM software version 4.11.2 (Gupta *et al*, 2007).

### Overlap between eY1H and ChIP interactions

ChIP interactions were downloaded from the modENCODE Project (http://www.modencode.org) on August 28, 2014. Comparison between ChIP and eY1H interactions was limited to 46 TFs that were both detected by eY1H and tested by ChIP. ChIP peaks were assigned to a promoter if the midpoint of the peak was located within the promoter sequence. Multiple ChIP peaks mapping to the same promoter were considered as one promoter–TF interaction. To determine the statistical significance of the overlap between the ChIP and eY1H datasets, we compared the ChIP data to 20,000 randomizations of the eY1H network, as described above.

### Gene expression level

Gene expression levels across development (from early embryo to young adult) were obtained from WormBase (project ID: PRJNA33023). To compare expression levels between TFs detected by different PDI mapping methods, we considered the maximum expression level across development to account for TFs expressed at one or few developmental stages. Tissue expression data for embryo (intestine, pan neuronal, pharyngeal muscle, body wall muscle, coelomocytes, hypodermis) and larvae (intestine, panneuronal, body wall muscle, coelomocytes, hypodermis) were obtained from Spencer *et al* (2011).

### Spatiotemporal overlap between TFs and genes

For each TF-gene pair, we calculated the Pearson correlation coefficient (PCC) between their tissue expression levels during embryo and larval stages (see previous section). TF-gene pairs with a PCC value above the 75$^{th}$ percentile of all PCC values (PCC ≥ 0.29) were considered as having overlapping expression patterns across tissues. The odds ratio was calculated as the ratio of interacting TF-gene pairs with overlapping expression patterns (compared to TF-gene pairs below the 75$^{th}$ percentile threshold) relative to that of non-interacting TF-gene pairs. For each predicted activator and repressor, we calculated the odds ratio and considered a TF to have spatiotemporal overlap with its targets if the odds ratio was above 1, and to have spatiotemporal overlap with its non-targets if the odds ratio was below 1.

### Co-expression network

Co-expression scores were determined for all pairs of *C. elegans* genes by integrating 123 expression profiling datasets as previously described (Chikina *et al*, 2009; Reece-Hoyes *et al*, 2013). The 101 datasets with a minimum of four experiments were defined as high-confidence. Genes present in fewer than 25 high-confidence expression datasets were removed from the analyses. To reduce the impact of noise in gene expression, genes that ranked in the lowest 10% expression level in more than half of the high-confidence expression profiling datasets were removed from the analyses.

### Transcriptional activator and repressor predictions

For this analysis, we considered 153 TFs that are present in the co-expression network and that bind at least 10 promoters in eY1H assays for which we have co-expression data. For each TF, we compared the co-expression scores of the TF and its eY1H targets with the co-expression scores of the TF and its eY1H non-targets. TFs that exhibit significantly higher co-expression with its eY1H targets than with non-targets (Mann–Whitney *U*-test) were predicted to be activators, while TFs that exhibit significantly higher co-expression with their non-targets were predicted to be repressors. To account for the number of TF activators and TF repressors expected by chance, we built a random model in which, for each TF, we shuffled its co-expression scores with the target and non-target genes and obtained predictions for potential activators or repressors in the randomized network. The randomization was performed 1,000 times.

### *C. elegans* strains

*C. elegans* strains were cultured and maintained by standard protocols (Brenner, 1974). N2 (Bristol) was used as the wild-type strain. The *nhr-102(tm1542)*, *nhr-142(tm4401)*, *nhr-273(tm1787)*, *nhr-42 (tm1375)*, *ceh-1(tm455)*, *ceh-38(tm321)*, *ceh-39(tm1336)*, and *C27F2.1(tm6453)* mutant strains were kindly provided by the National BioResource Project, Japan. The *nhr-66(ok940)*, *K08D12.2 (ok1863)*, *ceh-17(np1)*, *nhr-178(gk1005)*, *daf-19(m86);daf-12 (sa204)*, and the *daf-12(sa204)* mutant strains were obtained from the *C. elegans* Genetics Center (CGC). Mutant strains were genotyped and outcrossed four times with N2 (Dataset EV9).

### Real-time quantitative PCR

To measure the expression of DAF-19 eY1H targets, threefold embryos from *daf-19(m86);daf-12(sa204)* and *daf-12(sa204)* strains

were collected as previously described (Phirke *et al*, 2011). To measure the expression of targets of CEH-1, CEH-17, NHR-102 and NHR-142, N2, *ceh-1(tm455)*, *ceh-17(np1)*, *nhr-102(tm1542)*, *nhr-142(tm4401)* and *nhr-102(tm1542);nhr-142(tm4401)* strains were synchronized at L1 by incubating eggs in M9 media for 18 h. To measure the expression of targets of CEH-38 and CEH-39 synchronized N2, *ceh-38(tm321)* and *ceh-39(tm1336)* animals were harvested at the L4 stage. Total RNA was isolated using TRIzol (Invitrogen) and then purified using Direct-zol RNA MiniPrep kit (Zymo Research) including the DNAse I treatment step to remove contaminating DNA. cDNA was reverse-transcribed from RNA using oligo(dT)12–18 primer and M-MuLV reverse transcriptase (NEB). Primer sequences for real-time quantitative PCR were generated using the Roche and ThermoFisher Scientific primer design tools so that primers are located in different exons or in exon–exon junctions (Dataset EV9). Quantitative PCRs were performed in three technical replicates using the Applied Biosystems StepOnePlus Real-Time PCR System and Fast SYBR Green Master Mix (ThermoFisher Scientific). The $\Delta\Delta C_t$ method was used to determine the relative transcript abundance and was normalized to averaged *ama-1* and *act-1* mRNA levels (Livak & Schmittgen, 2001).

## Chemotaxis assays

Chemotaxis was assayed in 10-cm NGM-agar plates without peptone and cholesterol as previously described (Bargmann *et al*, 1993) with minor modifications. Briefly, 1 μl of chemoattractant (1:100 isoamyl alcohol in ethanol) and 1 μl of ethanol (control) were added on the agar on opposite sides of the plate, 1 cm from the edge of the plate. To facilitate counting, 1 μl of 10% azide was added to the chemoattractant and control spots to anesthetize the animals. Adult animals (100–200) were placed in the center of the plate. Plates were sealed with parafilm and incubated for 2 h at room temperature. Then, the number of animals within a radius of 1.1 cm of the chemoattractant (X) or the control spot (Y) was counted. The chemotaxis index was calculated as $(X - Y)/N$, where N is the number of animals seeded on the plate.

## Osmotic avoidance assays

Osmotic avoidance was assayed in 3.5-cm NGM-agar plates without peptone and cholesterol as previously described (Culotti & Russell, 1978) with minor modifications. Briefly, 15 μl of 40% glycerol was added to the center of the plate and plates were dried for 15 min at room temperature. Adult animals (50–200) were placed in the center of the plate and incubated for 60 min at room temperature. Then, the number of animals within a radius of 0.8 cm from the center of the plate was determined (I). The percentage of non-avoider animals was calculated as $I/N \times 100$, where N is the number of animals seeded on the plate.

## Gene ontology associations

For each TF, we performed a gene ontology (GO) enrichment analysis on its eY1H targets. TFs with a low out-degree (< 10) were removed from the analysis as they are unlikely to provide significant enrichments. TFs with a high out-degree (> 150) were also removed to avoid non-specific associations. Gene ontology biological process terms with fewer than 10 or more than 200 eY1H targets were removed to avoid very specific and non-specific terms. If several GO terms were annotated to the exact same set of target promoters, a single term was randomly selected and kept. The rest were removed from our test evaluation set. Only 102 non-redundant medium-sized biological process terms were evaluated for enrichment. Enrichment significance was calculated by a Fisher's exact test. We identified 84 "TF-GO term" associations with a *P*-value < 0.001. To account for the expected number of associations by chance, we repeated the same procedure on 1,000 randomizations of the PDI network.

## Toxicity assays

*E. coli* HT115 containing relevant RNAi plasmids (MacNeil *et al*, 2015) were inoculated into LB + 50 μg/ml ampicillin in 96-well deep-well plates and grown overnight at 37°C. The following day, fresh cultures were inoculated in 96-well deep-well plates with 100 μl of overnight culture in 1 ml of LB + 50 μg/ml ampicillin + 2 mM isopropyl β-D-thiogalactopyranoside (IPTG). Cultures were grown for 6 h (to an $OD_{600}$ of approximately 1), and bacteria were pelleted and resuspended in 250 μl of liquid NGM + 50 μg/ml ampicillin + 1 mM IPTG media. Synchronized L1 larvae were added to a final density of 200 animals/ml. 80 μl of *C. elegans*/bacteria suspension was added to 96-well flat-bottom plates containing 20 μl of the relevant xenobiotic (Dataset EV7). Plates were sealed with breathable films, covered with a plastic lid, and incubated at 20°C with agitation (100 rpm) for 3–4 days. Pictures were obtained using an EVOS FL Auto Cell Imaging System microscope (Invitrogen). Toxicity was considered positive when the animals developed slower or were sick or dead in two or more of four replicate experiments. Toxic TF-xenobiotic associations were tested using TF mutant strains.

## TF essentiality

Phenotypic data was obtained from WormBase version WS252 on March 22, 2016. Genes annotated as being lethal at any developmental stage either in mutants or by RNAi experiments were considered to be essential.

## Developmental rate assay

Animals were grown at 20°C. To measure the developmental rate, animals were first synchronized by L1 arrest. Briefly, eggs were collected by bleaching, washed four times in M9 buffer, and allowed to hatch in M9 buffer for 18 h. Following synchronization, animals were transferred to nematode growth media (NGM) plates containing *E. coli* OP50 and incubated at 20°C. After 45 h animals were washed off of the plates, mounted on agarose pads, and examined on a compound microscope. Animals were visually categorized into age groups based on the development of the vulva (MacNeil *et al*, 2013). At least 100 animals were scored per strain/RNAi treatment.

**Expanded View** for this article is available online.

## Acknowledgements

We thank members of the Walhout Lab for discussions and critical reading of the manuscript. We thank Alos Diallo for assistance in image processing

analyses. Strains were provided by the *C. elegans* Genetics Center (CGC), which is funded by NIH Office of Research Infrastructure Programs (P40 OD010440); and by the National BioResource Project, Japan. This work was supported by US National Institutes of Health grant GM082971 to AJMW and CLM.

## Author contributions

JIFB and AJMW conceived the project. JSR-H and SS performed eY1H experiments. JIFB and CP performed the data analysis with assistance from AM. JIFB and LK performed *C. elegans* experiments. JIFB, ADH, and AJMW generated the wTF3.0 list. CP was supervised by CLM. JIFB and AJMW wrote the manuscript with contributions from the other authors.

## Conflict of interest

The authors declare that they have no conflict of interest.

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
