## [Review Process File · Molecular Systems Biology]

A gene-centered *C. elegans* protein-DNA interaction network provides a backbone for functional predictions

Juan Fuxman Bass, Carles Pons, Lucie Kozlowski, John Reece-Hoyes, Shaleen Shrestha, Amy Holdorf, Akihiro Mori, Chad Myers and Albertha Walhout

Corresponding author: Albertha Walhout, University of Massachusetts Medical School

Review timeline:

Submission date:	13 June 2016
Editorial Decision:	14 July 2016
Revision received:	12 September 2016
Editorial Decision:	26 September 2016
Revision received:	27 September 2016
Accepted:	28 September 2016

Editor: Maria Polychronidou

Transaction Report:

1st Editorial Decision

14 July 2016

Thank you again for submitting your work to Molecular Systems Biology. We have now heard back from two of the three referees who agreed to evaluate your study. Reviewer #2 has informed us that s/he is not going to be able to provide comments in a timely manner, so in the interest of time, I prefer to make a decision based on the two available reports. As you will see below, the reviewers acknowledge that the study presents a potentially useful resource for the *C. elegans* community. However, they raise a number of concerns, which should be carefully addressed in a revision of the manuscript. The recommendations of the referees are clear, so there is no need to repeat the points listed below.

REFeree REPORTS

Reviewer #1:

Bass et al analyze *C. elegans* protein-DNA interactions using a heterologous yeast one hybrid system and a large collection of transcription factors and promoters. A total of approx 10% of all such interactions are mapped in the study, representing the largest single effort in this area. The authors proceed to validate their dataset bioinformatically, making many interesting predictions of protein function. They also validate some predictions experimentally, perhaps most interestingly they find that many *C. elegans* nuclear hormone receptors are required for efficient xenobiotic response.

The manuscript is clearly written and informative, and the conclusions are justified by the evidence.

The bioinformatic methods used are well applied, and the biological findings are interesting, albeit somewhat preliminary. The main strength of the work is the provided resource that complements work based on other methods, and allows systems level analyses of gene regulation in *C. elegans*.

Specific points

- Despite the large dataset, the authors do not make any effort to analyze sequence-function relationships. Such analyses should be included, even if the analyses fail to differentiate between Y1H active elements that have ChIPseq peaks or motifs and active elements that do not etc.
- Authors should indicate effect sizes in addition to the p-values in the text. Also, the focus is on predictions based on aggregate data, and it is not clear what is the confidence of individual edges. This could be estimated and discussed.
- p4 top authors discuss that presence of binding site is poor predictor of binding in vivo. However, binding is also poor predictor of transcriptional activity. Does binding in Y1H equal activity? To what extent does Y1H help to predict which binding events are activating? The authors should comment on this.
- p4 bottom authors should discuss the area more broadly, including the work in sea urchin to map regulatory interactions, and new massively parallel reporter assays etc
- p8 many promoters were removed as they conferred high background. The authors should analyze the sequences, and test whether these contain binding sites for yeast factors etc. Also, can the method not be modified to detect repressors?
- p14 authors validate findings using RT-PCR. It is unclear whether the target genes tested were included in the original discovery set.
- p17 clarify the concepts dauer constitutive and dauer defective
- p20-21 authors argue that factors with different targets are essential because of lack of redundancy. But their example is confusing and not consistent with their earlier idea that nuclear hormone receptors respond to environment (and thus allow different inputs to regulate the same genes). Both inputs and outputs need to be equivalent in order for something to be redundant.
- p24 last sentence of first paragraph is incorrect, bifunctional factors can be detected with less sensitivity
- some figure panels (1a, 1c, 3a, 7b) are uninformative, or contain excess data that is not central for understanding of the point.

Reviewer #3:

Summary and general assessment:

This report from the Walhout group focuses on an extensive analysis of transcription factor-promoter interactions in *C. elegans*, leading to the identification of several thousand of such interactions via the yeast one-hybrid system. Although this is an in vitro assay, the authors validate the relevancy of their findings by comparison with independent large scale datasets such as ChIPseq and gene expression. They build a network of these interactions, and assign activator or repressor functions to a subset, using the yeast one-hybrid data in combination with other existing datasets. The authors then test whether target genes share functions, whether related TFs show more similar patterns, etc. Overall, this analysis provides a substantial new resource of information for others to mine for potential regulatory interactions and greatly expands our first-level understanding of TF binding to candidate promoter sequences. The analysis is generally well done (see concerns below), and the data is presented in such a way as to facilitate others utilizing it.

Major points:

Why were so few regulatory interactions predicted from the analysis described in Figure 3b - 44/153? This seems low and is never explained by the authors. They simply say it's more than random expectation. It is not surprising that they were able to achieve greater than random, but that is not sufficient to say this was a robust analysis. Is there a reason why so many factors did not yield predictive interactions from the y1h? The data provided do not allow for a good judgment regarding the reliability and robustness of the network. Moreover, the authors frequently use percentages throughout the subsequent analysis of this PDI network, but by the time 44 factors are parsed out, the actual N is often very low. This also leads to some worry that the interaction data is less robust than one would hope.

The effects on target gene regulation are underwhelming for activators and repressors, and it is not really clear why this is so. The authors list some possibilities in the discussion, but no obvious explanation why so few genes change expression - perhaps it is the stage at which the RNA was collected or the fact that it was collected from whole animals, missing or masking the regulatory event. Moreover, the methods describe different stages of development selected for different RNA analyses without really explaining why such stages were chosen. For example, when looking at *daf-19* targets, they looked in embryos when the phenotype is a larval dauer phenotype. This all should be made clearer.

Minor points:

Intro line 2: comma after "stresses" (necessary because you are listing phrases, not just words)

Intro p5: The sentences that equate compact genome with "few distal enhancers" should be re-written. A compact genome does not "therefore" rule out the possibility of distal enhancers. No one has actually systematically, experimentally, looked for distal enhancers in *C. elegans*, so it is difficult to say whether they are rare, non-existent or frequent.

Results p7: Although previously published, perhaps a sentence about the genes represented in the promoterome would be useful to the reader - do they tend to encode particular functions? chromosome locations?

Results p10: Reference(s) needed for the "compendium of co-expression data from 123...datasets"

Results p12: The analysis in which activators and repressors are shown to have more similar expression patterns with their targets than non-targets (Fig 3F) seems problematic, as there are so few targets (usually around 10 or so) than non-targets (~3000). Therefore the comparison between targets and nontargets is extremely lopsided, and there could be many reasons why the non-targets have less correlated expression.

Results p 21, and again in Discussion, p 24: "identify TF pairs that bind overlapping sets of targets in eY1H assays and that, therefore, likely bind similar DNA sequences". It is not clear why factors that bind at the same genes likely bind similar consensus elements. It's entirely possible that they bind unrelated sequences to coordinately regulate target gene expression. This statement seems conflated.

I do not understand the methods for the coexpression network. I read this as starting with a set of 123 global gene expression experiments, some of which were removed for various reasons. Why would such a gene expression experiment only have 5 genes in it, or need at least 4 experiments? I don't follow at all what is meant here.

Why use 10 promoter binding events as the cutoff for inclusion of a TF in the network? What happens if this cutoff is raised or lowered? The authors need to provide justification for this choice in boundary.

The terms "out degree" or "in degree" in figure 1c are unnecessarily confusing. The legend provides an explanation but it is not easy to figure out why these phrases are used to denote the number of promoters bound by a TF, and the number of TFs that bind a given promoter. This seems like excess jargon.

1st Revision - authors' response

12 September 2016

We thank the reviewers and Editor for their helpful comments and suggestions that helped us to further improve our manuscript.

Reviewer #1:

Bass et al analyze *C. elegans* protein-DNA interactions using a heterologous yeast one hybrid system and a large collection of transcription factors and promoters. A total of approx 10% of all such interactions are mapped in the study, representing the largest single effort in this area. The authors proceed to validate their dataset bioinformatically, making many interesting predictions of protein function. They also validate some predictions experimentally, perhaps most interestingly they find that many *C. elegans* nuclear hormone receptors are required for efficient xenobiotic response.

The manuscript is clearly written and informative, and the conclusions are justified by the evidence. The bioinformatic methods used are well applied, and the biological findings are interesting, albeit somewhat preliminary. The main strength of the work is the provided resource that complements work based on other methods, and allows systems level analyses of gene regulation in *C. elegans*.

Specific points

- Despite the large dataset, the authors do not make any effort to analyze sequence-function relationships. Such analyses should be included, even if the analyses fail to differentiate between Y1H active elements that have ChIPseq peaks or motifs and active elements that do not etc.

*To further show the quality of our PDI network, we evaluated motif enrichments in the promoter sequences of eY1H positive targets using CisFinder (PMID: 19740934). For each TF, the 500 bp proximal promoter sequences of eY1H targets was used to identify motifs enriched compared to the sequences for eY1H non-targets (promoters not bound by that TF, but present in the overall network). Elementary motifs were obtained using an FDR of 0.05 and a motif enrichment greater than 2. Motifs were determined for the 77 TFs that have 50 or more eY1H targets in the PDI network to avoid obtaining low-quality motifs. These motifs are included in **Dataset EV2**. For the 25 TFs for which PBM data was available we compared the motif logo derived from eY1H data with the one derived from PBMs. Statistical significance between the eY1H- and PBM- derived motifs was determined using the TOMTOM software version 4.11.2 (PMID: 17324271). Twenty-one of 25 motifs were statistically significant showing that eY1H can be used for de novo motif discovery (shown in **Figure EV1**). These results are discussed in the revised manuscript (page 9 bottom, methods, **Figure EV1** and **Dataset EV2**).*

Comparison of PBM-derived motif enrichment between eY1H and ChIP data has been recently reported (PMID: 25905672). This study, which was performed using the same eY1H dataset included here, found that motifs are more frequently enriched in eY1H data than in ChIP-seq data, further supporting the quality of our dataset.

*Comparison of de novo motif predictions with ChIP data was more challenging since there was only one TF (LSY-2) with enough eY1H targets to perform the enrichment analyses, and with PBM data with which to compare our predictions. Overall, the motifs predicted based on eY1H positive sequences (both positive or negative for ChIP interactions) was similar to the motif derived from PBM data (see **Rebuttal Figure 1**). Motif prediction using ChIP positive and eY1H negative interactions did not match the reported motif. This further supports the high quality of the eY1H data. However, given that these results are based on only one TF, we did not include them in the revised manuscript.*

Rebuttal Figure 1 – Comparison between motifs derived from eY1H and ChIP sequences that interact with LSY-2.

The top two scoring motifs derived from eY1H+ ChIP -, eY1H+ ChIP+ and eY1H - ChIP+ data are compared to the motif logo derived from PBM data reported in CisBP (PMID: 25215497). The Venn diagram shows the number of targets interacting with LSY-2 by eY1H and ChIP.

- Authors should indicate effect sizes in addition to the p-values in the text. Also, the focus is on predictions based on aggregate data, and it is not clear what is the confidence of individual edges. This could be estimated and discussed.

P-values are indicated in the figures and referenced in figure legends. Effect sizes can be estimated in each of the figures. Although it would be of great use, quantitative confidence on individual edges cannot be estimated with the current knowledge in the field as it depends on multiple variables whose relative contribution have not been determined yet.

- p4 top authors discuss that presence of binding site is poor predictor of binding in vivo. However, binding is also poor predictor of transcriptional activity. Does binding in Y1H equal activity? To what extent does Y1H help to predict which binding events are activating? The authors should comment on this.

*The reviewer is right to point out that Y1H binding does not necessarily imply transcriptional activity. In a previous work, we found that several eY1H interactions are regulatory but others are not, at least in the one tissue / condition and stage tested (PMID: 26430702). We added a sentence in the introduction (page 5, bottom): “Further, we have found that eY1H interactions could be validated in animals harboring transcriptional fusion reporter constructs fed with bacteria expressing RNAi clones against different TFs”. Further, in the present manuscript we found that 30-60% of interactions tested in TF mutant animals (9-34 targets per TF tested in seven TF mutant animals) are regulatory, even when a single stage and condition was evaluated. This is discussed in **Figures 3G, 5B and 8B**.*

- p4 bottom authors should discuss the area more broadly, including the work in sea urchin to map regulatory interactions, and new massively parallel reporter assays etc

A discussion on regulatory interaction mapping in different organisms, and on MPRAs have been added to the introduction (page 3):

*“Gene regulatory networks have been extensively studied at a small or medium scale in different metazoan organisms including nematodes, sea urchins, fruit flies, and mammals. For instance, GRNs regulating development in the sea urchin *Strongylocentrotus purpuratus* embryo have been delineated based on spatiotemporal expression patterns of TFs and signaling molecules, ChIP-seq data and gene knockdown (reviewed in 26962438). Similarly, developmental GRNs have been studied in *Drosophila melanogaster* using genetic perturbations, expression profiling, transgenic reporters, ChIP-seq data, and mathematical modeling (reviewed in 19874814). More recently, massively parallel reporter assays have been used for an in depth study of the function of regulatory sequences in cell lines, mouse tissues, yeast and bacteria (reviewed in 26072432).”*

- p8 many promoters were removed as they conferred high background. The authors should analyze the sequences, and test whether these contain binding sites for yeast factors etc. Also, can the method not be modified to detect repressors?

eYIH assays can detect both activators and repressors; it solely detects binding. The assay is based on binding of a TF that is fused to the strong yeast Gal4p transcriptional activation domain. Thus, even repressors will activate in yeast in this context. We agree that high background may be caused by yeast TFs binding and activating some promoters. However, a comprehensive analysis of yeast TF binding sites would require extensive effort (especially the follow-up testing – which would be needed to added meaning to the analysis) is beyond the scope of the paper.

- p14 authors validate findings using RT-PCR. It is unclear whether the target genes tested were included in the original discovery set.

All the targets tested by RT-qPCR were identified in the eYIH dataset. This is mentioned in the text in each example. For instance: “We evaluated the expression of ten CEH-17 eYIH targets...”

- p17 clarify the concepts dauer constitutive and dauer defective

Dauer constitutive: A mutant phenotype where animals form dauer larvae under conditions that would not induce dauer formation in wild type animals.

Dauer defective: A mutant phenotype where animals are unable to form dauer larva under conditions that would induce dauer formation in wild type animals.

Given that these phenotypes are well known in the C. elegans community, we have not included the definitions in the manuscript.

- p20-21 authors argue that factors with different targets are essential because of lack of redundancy. But their example is confusing and not consistent with their earlier idea that nuclear hormone receptors respond to environment (and thus allow different inputs to regulate the same genes). Both inputs and outputs need to be equivalent in order for something to be redundant.

We agree with the reviewer that redundancy in binding does not necessarily imply redundancy in function (regulation by different inputs, in different tissues/stages or different regulatory activity influence redundancy in function). As such, each edge in the association network should not be interpreted as redundancy between the TFs involved. However, when analyzed globally we detected an enrichment for non-essential TF among those that are connected to 2+ other TFs validating our overall conclusions based on the aggregated data. The limitations of this analysis are discussed in page 26:

“In addition, many TF-pairs connected in the association network could have unrelated functions if they are expressed in different tissues/conditions/stages, and/or respond to different environmental cues as we have observed for NHR mutant animals exposed to different xenobiotics. Overall, these observations highlight the complexity in C. elegans gene regulation and the importance of integrating multiple functional datasets.”

- p24 last sentence of first paragraph is incorrect, bifunctional factors can be detected with less sensitivity

The analyses performed to predict regulatory functions assign as activator to TFs that have higher coexpression with targets than with non-targets and vice versa for repressors. Given this binary assignment we would miss bifunctional TFs. We modified this sentence to clarify this point:

“Finally, given that predictions are based on the comparison between TF coexpression with targets and non-targets (higher or lower), we cannot identify bifunctional TFs using this approach. Therefore, TFs that can function as both activators and repressors would have been missed in this analysis.”

- some figure panels (1a, 1c, 3a, 7b) are uninformative, or contain excess data that is not central for understanding of the point.

We respectfully disagree and do consider these figures to be informative and relevant for the paper.

1A: Although some of these numbers are referenced in the text, the flowchart helps understand where the data is coming from.

1C: This figure condenses information on the distribution of the interactions detected as well as TF and promoter connectivity. We believe that presenting network structure will be useful for the systems biology community studying biological networks.

3A: This figure, although it does not convey information, briefly presents the rationale for the method employed in the entire figure.

*7B: This figure presents the data from which aggregated analyses in **Figures 7C-E** are based. In addition, the expanded view of this figure (**Figure EV3**) show TF-pairs that may be functionally related (redundant, antagonistic, etc.) that could be explored by the *C. elegans* community for follow up experiments.*

Reviewer #3:

Summary and general assessment:

This report from the Walhout group focuses on an extensive analysis of transcription factor-promoter interactions in *C. elegans*, leading to the identification of several thousand of such interactions via the yeast one-hybrid system. Although this is an in vitro assay, the authors validate the relevancy of their findings by comparison with independent large scale datasets such as ChIPseq and gene expression. They build a network of these interactions, and assign activator or repressor functions to a subset, using the yeast one-hybrid data in combination with other existing datasets. The authors then test whether target genes share functions, whether related TFs show more similar patterns, etc. Overall, this analysis provides a substantial new resource of information for others to mine for potential regulatory interactions and greatly expands our first-level understanding of TF binding to candidate promoter sequences. The analysis is generally well done (see concerns below), and the data is presented in such a way as to facilitate others utilizing it.

Major points:

Why were so few regulatory interactions predicted from the analysis described in Figure 3b - 44/153? This seems low and is never explained by the authors. They simply say it's more than random expectation. It is not surprising that they were able to achieve greater than random, but that is not sufficient to say this was a robust analysis. Is there a reason why so many factors did not yield predictive interactions from the y1h? The data provided do not allow for a good judgment regarding the reliability and robustness of the network. Moreover, the authors frequently use percentages throughout the subsequent analysis of this PDI network, but by the time 44 factors are parsed out, the actual N is often very low. This also leads to some worry that the interaction data is less robust than one would hope.

We agree with the reviewer that although the number of predictions is encouraging we were not able to provide predictions for many TFs. We thoroughly discuss this in the Discussion section:

“For instance, by integrating the PDI network with a compendium of expression profiling data we provide predictions for the regulatory functions of 44 TFs, which are largely consistent with published regulatory functions and those predicted based on TF-cofactor interactions. This number of predictions is encouraging, considering that there are several reasons that can explain lack of correlation between TF and target gene expression. First, our predictions are based on the assumption that there is a strong correlation between TF mRNA expression and TF activity. However, this may not be the case for many TFs whose regulatory function is modulated by ligand binding, post-translational modifications or dimerization with other factors (Walhout, 2011). For instance, although we provide regulatory predictions for 29% of the TFs evaluated, we did so only for 15% of the NHRs tested, a family known to be modulated by ligands and that frequently forms heterodimers. Second, predictions for some TFs may be missed due to redundancy as changes in the expression of a single TF may not be sufficient to affect the expression of its target gene. Finally, given that predictions are based on the comparison between TF co-expression with targets and non-

targets (higher or lower), we cannot identify bifunctional TFs using this approach. Therefore, so many TFs that can function as both activators and repressors would have been missed in this analysis.”

The main reason for not mentioning this in Results is that the interpretation is supported by results discussed in later figures in the paper.

Given the comments above, the fact that we did not provide functional predictions for many TFs does not stem from a lack of robustness in the dataset but from the complexity of functional relationships. Indeed, the high quality of the dataset is illustrated by the overlap with existing datasets and the multiple functional validations performed in **Figures 3, 5, 6 and 8**.

The effects on target gene regulation are underwhelming for activators and repressors, and it is not really clear why this is so. The authors list some possibilities in the discussion, but no obvious explanation why so few genes change expression - perhaps it is the stage at which the RNA was collected or the fact that it was collected from whole animals, missing or masking the regulatory event.

We were able to ‘validate’ 30-60% of the interactions tested in TF mutant strains. We consider this number to be encouraging for multiple reasons:

- Functional validations were performed in a single stage and condition and using whole animals. Therefore, testing more conditions/stages or measuring expression changes in tissues (rather than in whole animals) will likely ‘validate’ more of the interactions.
- Redundancy with different TFs may mask the effect of a particular TF in gene regulation.
- Not all TF binding is regulatory in vivo, even for interactions detected by ChIP, as previously reported (PMID: 26430702).

We agree with the reviewer about discussing these points, which are now in the Discussion section (page 24, bottom).

Moreover, the methods describe different stages of development selected for different RNA analyses without really explaining why such stages were chosen. For example, when looking at *daf-19* targets, they looked in embryos when the phenotype is a larval dauer phenotype. This all should be made clearer.

Stages were chosen based on when TFs and target genes are expressed. For example, *daf-19* is most highly expressed at late stages in embryo development and studies looking at targets of *DAF-19* were performed at this stage (PMID: 21740898). Developmental defects occurring at embryonic stages can result in phenotypes that are evidenced at later stages (chemotaxis, dauer, osmotic avoidance). This is now clarified on page 18 of the revised manuscript.

Minor points:

Intro line 2: comma after "stresses" (necessary because you are listing phrases, not just words)

Corrected.

Intro p5: The sentences that equate compact genome with "few distal enhancers" should be rewritten. A compact genome does not "therefore" rule out the possibility of distal enhancers. No one has actually systematically, experimentally, looked for distal enhancers in *C. elegans*, so it is difficult to say whether they are rare, non-existent or frequent.

We removed the reference to distal enhancers. The sentence now reads as: “Therefore, most gene regulation likely occurs through proximal gene promoters.”

Results p7: Although previously published, perhaps a sentence about the genes represented in the promoterome would be useful to the reader - do they tend to encode particular functions? chromosome locations?

The following sentence was added in page 8: “This collection comprises ~5,500 promoter regions of 0.3-2 kb of genes encoding for different molecular functions, and that are distributed across all *C.*”

elegans chromosomes."

Results p10: Reference(s) needed for the "compendium of co-expression data from 123...datasets"
The reference was added to the results and methods sections in the revised manuscript.

Results p12: The analysis in which activators and repressors are shown to have more similar expression patterns with their targets than non-targets (Fig 3F) seems problematic, as there are so few targets (usually around 10 or so) than non-targets (~3000). Therefore, the comparison between targets and nontargets is extremely lopsided, and there could be many reasons why the non-targets have less correlated expression.

*For the results in **Figure 3F**, as described in the methods section, we measured an odds ratio, which accounts for the total number of targets and nontargets. More specifically, for each TF, we measured the following:*

OR = (# of targets with expression similarity > threshold / total # of targets) / (# of non-targets with expression similarity > threshold / total # of non-targets)

*Note that the number of targets and non-targets appears in the numerator and denominator. For the plot in **Figure 3F**, we then simply counted for each predicted activator or repressor TF, how many of these OR were < or > 1. Statistical significance was measured with a Fisher exact test.*

Details about this analysis are provided in the figure legend and in the Methods section.

Results p 21, and again in Discussion, p 24: "identify TF pairs that bind overlapping sets of targets in eYIH assays and that, therefore, likely bind similar DNA sequences". It is not clear why factors that bind at the same genes likely bind similar consensus elements. It's entirely possible that they bind unrelated sequences to coordinately regulate target gene expression. This statement seems conflated.

The reviewer raised a valid point. Although in general we have found that TFs that bind to overlapping sets of genes across the network (Jaccard > 0.2) tend to have a high percentage of amino acid identity in their DNA binding domains (PMID: 23791784) and bind to similar motifs (PMID: 25910213), this may not always be the case. We made this point more explicit in the revised manuscript (page 22):

"To globally uncover functional relationships between TFs, we leveraged the PDI network to identify TF pairs that bind overlapping sets of targets in eYIH assays and that, as we have previously shown, frequently bind to similar DNA sequences."

I do not understand the methods for the coexpression network. I read this as starting with a set of 123 global gene expression experiments, some of which were removed for various reasons. Why would such a gene expression experiment only have 5 genes in it, or need at least 4 experiments? I don't follow at all what is meant here.

We thank the reviewer for pointing this out. Pearson correlation, the metric we used for measuring similarity of gene pairs' expression profiles is not robust when computed over too short a profile (i.e. the number of expression datasets). As some expression datasets have very few samples, we required genes to appear in at least 25 of the 101 expression datasets with 4 or more samples in each, which we defined as high-confidence expression datasets. Genes whose co-expression values were calculated with fewer high-confidence expression datasets were not used in our analysis. We agree that the "5 genes" statement is confusing-- we had a small number of datasets for which most of the genes were filtered for technical reasons. These datasets are of course not informative and did not contain information for any of our genes in the PDI network. We have removed this statement and just indicate that the 101 datasets with a minimum of four experiments and expression measurements for genes in our PDI network were used for the co-expression analysis.

Why use 10 promoter binding events as the cutoff for inclusion of a TF in the network? What happens if this cutoff is raised or lowered? The authors need to provide justification for this choice in boundary.

The choice of boundary is based on sample size to be able to detect statistically significant differences between co-expression with targets and non-targets. In the revised manuscript we included an analysis changing that boundary to 5, 8, 15 and 20 (Figure 3B, bottom), obtaining similar results compared to the boundary of 10 PDIs.

The terms "out degree" or "in degree" in figure 1c are unnecessarily confusing. The legend provides an explanation but it is not easy to figure out why these phrases are used to denote the number of promoters bound by a TF, and the number of TFs that bind a given promoter. This seems like excess jargon.

These are standard terms in the field. Having the explanation in the figure would make it harder to read. The definitions are included in the figure legend.

2nd Editorial Decision

26 September 2016

Thank you again for sending us your revised manuscript. We have now heard back from reviewer #1 and as you will see below, s/he thinks that the reviewers' concerns have been addressed. However, s/he requests adding a sentence in the discussion regarding the biological significance of the TF-promoter interactions (see comments below).

REFeree REPORTS

Reviewer #1:

The authors have addressed most of my criticisms. The remaining issue is the reliability of the individual predictions, which is also pointed out by the other referee. To address this, I would suggest that the authors add statement to the discussion that more explicitly states that the validation experiments establish that the Y1H network is enriched in biologically relevant interactions, but do not establish biological significance of any individual interaction outside those that are validated with the follow up experiments themselves.

2nd Revision - authors' response

27 September 2016

Response to Reviewer #1:

The authors have addressed most of my criticisms. The remaining issue is the reliability of the individual predictions, which is also pointed out by the other referee. To address this, I would suggest that the authors add statement to the discussion that more explicitly states that the validation experiments establish that the Y1H network is enriched in biologically relevant interactions, but do not establish biological significance of any individual interaction outside those that are validated with the follow up experiments themselves.

Thank you very much for the final comments. We respectfully argue that we have discussed at great length the topic raised once again, see the following in the Discussion:

Indeed, 30-60% of the PDIs tested in TF mutant animals resulted in changes in the expression of target genes. This result is similar to what has been previously reported for Y1H-derived interactions in Arabidopsis thaliana and Drosophila melanogaster (Brady et al, 2011; Hens et al, 2011). There are several reasons why we failed to detect a regulatory role for some PDIs detected by eY1H (Walkout, 2011). First, some PDIs identified in yeast may not occur in vivo. Second, some physical interactions, although occurring in vivo, may be neutral and have no regulatory consequence (MacNeil et al, 2015). Third, some regulatory interactions may have been missed as

they were tested in a single stage and condition, and because expression changes were measured in whole animals losing tissue resolution. Finally, some regulatory interactions may have been missed due to redundancy with other TFs.

We don't feel that stating another point as indicated by the reviewer will do justice to the paper, we are up front and honest and open throughout.

3rd Editorial Decision

28 September 2016

Thank you again for sending us your revised manuscript. We are now satisfied with the modifications made and I am pleased to inform you that your paper has been accepted for publication.

Corresponding Author Name: Albertha J.M. Walhout

Manuscript Number: 16-7131